# Rapid multi-directed cholinergic transmission in the central nervous system

Santhosh Sethuramanujam[1,9], Akihiro Matsumoto [2,9], Geoff deRosenroll[1], Benjamin Murphy-Baum[1], Claudio Grosman[3], J Michael McIntosh[4,5,6], Miao Jing[7], Yulong Li [7], David Berson[8], Keisuke Yonehara [2✉] & Gautam B. Awatramani [1✉]

In many parts of the central nervous system, including the retina, it is unclear whether cholinergic transmission is mediated by rapid, point-to-point synaptic mechanisms, or slower, broad-scale 'non-synaptic' mechanisms. Here, we characterized the ultrastructural features of cholinergic connections between direction-selective starburst amacrine cells and downstream ganglion cells in an existing serial electron microscopy data set, as well as their functional properties using electrophysiology and two-photon acetylcholine (ACh) imaging. Correlative results demonstrate that a 'tripartite' structure facilitates a 'multi-directed' form of transmission, in which ACh released from a single vesicle rapidly (~1 ms) co-activates receptors expressed in multiple neurons located within ~1 μm of the release site. Cholinergic signals are direction-selective at a local, but not global scale, and facilitate the transfer of information from starburst to ganglion cell dendrites. These results suggest a distinct operational framework for cholinergic signaling that bears the hallmarks of synaptic and non-synaptic forms of transmission.

[1] Department of Biology, University of Victoria, Victoria, BC, Canada. [2] Danish Research Institute of Translational Neuroscience – DANDRITE, Nordic-EMBL Partnership for Molecular Medicine, Department of Biomedicine, Aarhus University, Aarhus C, Denmark. [3] Department of Molecular and Integrative Physiology, 407 S. Goodwin Ave, Urbana, IL 61801, USA. [4] George E. Whalen Veterans Affairs Medical Center, Department of Psychiatry, School of Biological Sciences, University of Utah, Salt Lake City, UT, USA. [5] Department of Psychiatry; School of Biological Sciences, University of Utah, Salt Lake City, UT, USA. [6] School of Biological Sciences, University of Utah, Salt Lake City, UT, USA. [7] State Key Laboratory of Membrane Biology, Peking University School of Life Sciences, Beijing, China. [8] Neuroscience, Brown University, Providence, RI, USA. [9] These authors contributed equally: Santhosh Sethuramanujam, Akihiro Matsumoto. ✉email: keisuke.yonehara@dandrite.au.dk; gautam@uvic.ca

At central synapses, elaborate trans-cellular assemblies position postsynaptic receptors near presynaptic vesicle release sites (<0.3 µm), ensuring the delivery of high concentrations of neurotransmitters to those receptors[1–5], and rapid information transfer. But the effects of neurotransmitter release can extend beyond such point-to-point transmission with "spillover" or "volume transmission" to extrasynaptic sites. These "non-synaptic" actions are supported by distinct receptors with properties that appear well-matched to the slower kinetics and lower neurotransmitter concentrations experienced at extrasynaptic sites[6–9]. Importantly, the neurotransmitter concentrations at these distal sites reflect pooled contributions from many release sites[1,10–12], which reduces the spatial and temporal precision of neurotransmission.

Do cholinergic neurons employ the fast synaptic, point-to-point signaling that is typical of other fast neurotransmitters[13–19]? Or do they use the diffuse, and slow, extrasynaptic signaling that is commonplace for neuromodulators such as peptides and monoamines[20–24]? Analysis of central cholinergic synapses has proven to be more challenging than for most fast transmitter systems, and have left these questions unanswered. A major constraint has been that the structural descriptions of cholinergic synaptic microcircuits have not been complete enough to fully inform on the functional analysis. In the retina, however, this is changing because of striking advances in our understanding of cholinergic/GABAergic "starburst" amacrine cells. Detailed mapping of its circuits[25,26], now offers the opportunity to probe fundamental principles of cholinergic signal transmission.

The outputs of starburst amacrine cells to direction-selective ganglion cells (DSGCs) have been mapped with exquisite anatomical and functional detail (Fig. 1a, b). About decade ago, Briggman et al. (2011) reconstructed the starburst-DSGC circuitry using serial block-face electron microscopy (SBEM) and observed a prominent asymmetry in their synaptic connectivity[25] (Fig. 1b). They noted that the majority (~90%) of the large "wraparound" synaptic contacts made by radiating starburst dendrites arise from cells that have their somas displaced toward the "null-side" of the DSGC's receptive field (i.e., the side from which null stimuli first enter the receptive field; Fig. 1b). The skewed distribution of "null-connections" provides the structural basis for the asymmetric GABAergic inhibition observed in DSGCs, which endows them with direction selectivity[27–32] (Fig. 1b). On the other hand, stimulating starburst cells on any side of the DSGC evokes phasic cholinergic responses[27–31] (Fig. 1b). How symmetrical patterns of cholinergic excitation arise from an apparently asymmetric wiring pattern remains mysterious[25]. Solving this puzzle requires a detailed knowledge of the spatiotemporal scale over which individual ACh connections exerts their actions.

To this end, we examined the anatomical and functional properties of cholinergic signals occurring at individual starburst varicosities and present evidence for a "multi-directed" form of cholinergic transmission. Analysis of an existing SBEM data set[26] revealed "tripartite" complexes, in which a single starburst varicosity makes a classical 'wraparound' synapse as well as a "peripheral" contact with dendrites of distinct DSGCs. Electrophysiological and optical measurements show that ACh released from individual sites locally activate multiple postsynaptic DSGC dendrites with millisecond precision. Furthermore, cholinergic inputs to DSGC dendrites appear well-tuned for direction on the local, but not global scale. We discuss the biophysical basis for rapid, broad-scale ACh transmission, and how the tuning of ACh inputs could increase the efficiency of the direction-selective circuit.

## Results

### "Peripheral" contacts: points of cholinergic transmission? It has often been hypothesized that the symmetrical cholinergic

receptive fields of DSGCs is a consequence of volume transmission[25,29,31]. In this model, single starburst varicosities release ACh molecules that—unlike the GABA they also release—spread beyond the confines of the synapse to activate cholinergic receptors located on neighboring DSGC dendrites, over relatively slow time scales. However, cholinergic responses are strong and 'phasic', indicating that ACh diffuses over relatively short distances to activate target receptors[27–31]. To better understand the structure and function of cholinergic transmission, we first examined the abundance of secondary DSGC dendrites in the neighborhood of starburst-to-DSGC wraparound synapses.

In an existing SBEM stack[26] (50 × 210 × 260 µm³), we reconstructed 82 ON starburst cells in sufficient detail to infer the directional preferences of their dendrites based on the soma location (Fig. 1a, c; the radiating dendrites of starbursts are strongly activated by centrifugal motion, i.e., objects moving from the soma to the dendritic tips[33,34]). We also reconstructed dendrites arising from 32 ON-OFF DSGCs, of which only half had somas in the volume (Fig. 1a; Supplementary Fig. 1). As shown before[25], mapping the asymmetry of ON starburst synaptic contacts onto the DSGCs allowed us to divide the DSGCs into four groups, each encoding motion in one of the four cardinal directions (Fig. 1c; Supplementary Fig. 1b).

We mapped a small fraction of synapses ($n = 191$) onto six DSGCs with different directional preferences. Then, we measured the distance from each of these synapses to the closest point on the surface of any other ON-OFF DSGC. In nearly a third of the cases, the varicosity was in direct contact with the dendrite of another DSGC (Fig. 1d–f). In 77% of the total cases, there was a dendrite of another DSGC within 2 µm (Fig. 1f). The directional preference of the DSGC making such secondary contacts bore no consistent relationship to that of the DSGC receiving the synaptic contact (Fig. 1g). However, it was less common to find that both contacts shared the same preference, presumably because of the mosaic organization of dendritic fields of single DSGC subtypes.

The secondary "peripheral" contacts made by starburst varicosities were devoid of pre- or postsynaptic specializations, as with many cholinergic synapses elsewhere in the central nervous system[21,22,35]. The lack of well-defined ultrastructural elements made it difficult to verify that peripheral contacts truly represent additional points of cholinergic transmission (Fig. 1d, e). However, further correlative electrophysiological evidence demonstrated that ACh signals are multi-directed on a fine spatiotemporal scale (Fig. 2), indicating that transmission likely occurs at both contacts in these tripartite complexes.

**Rapid, multi-directed cholinergic transmission.** Dual voltage-clamp recordings from nearby DSGCs (inter-somatic distance <50 µm; Fig. 2a) revealed that a significant fraction of spontaneous cholinergic miniature excitatory postsynaptic currents (sEPSCs) occurred synchronously (Fig. 2b–d). Spontaneous inputs were measured at a holding potential of −60 mV (~$E_{Cl}$) in the presence of a drug cocktail containing 50 µM DL-AP4, 20 µM CNQX and 100 µM UBP-310, which block AMPA, NMDA and kainate-type glutamate receptors, as well as ON bipolar cell activity mediated by type 6 metabotropic glutamate receptors. Further pharmacological analysis revealed that sEPSCs are largely mediated by α6 subunit-containing receptors ($n = 5$; Supplementary Fig. 2), consistent with single cell transcriptomic profiling results that suggest a predominant expression of α6-nicotinic accetylcholine receptors (nAChRs) in DSGCs[36].

When the currents recorded in a given DSGC was averaged over a period specified by the occurrence of a sEPSC in a neighboring DSGC (i.e., the sEPSC-triggered average; STA), the resulting temporal waveform was indistinguishable from that of the average sEPSC (Fig. 2c, d). We estimated the magnitude of the

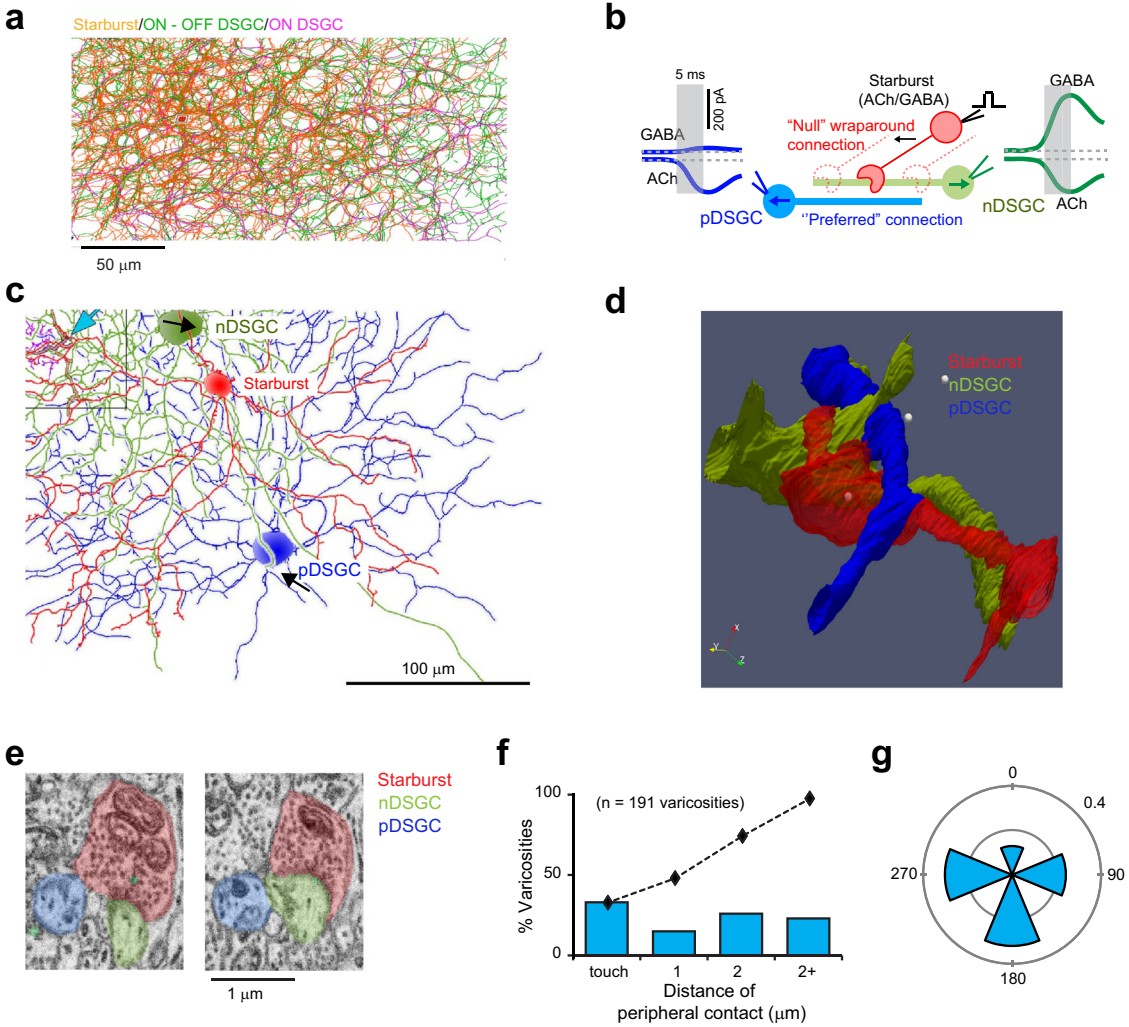

**Fig. 1 SBEM analysis reveals putative sites of cholinergic transmission. a** A flat-mount view of a small section of the inner-plexiform layer in the retina showing the dendrites of 82 ON GABAergic/cholinergic starburst amacrine cells and 32 ON-OFF DSGCs (both ON and ON-OFF DSGCs are shown) reconstructed from an existing SBEM dataset[26]. These partially reconstructed dendrites show the typical "honeycomb" plexus formed by the high level of co-fasciculation in the starburst/DSGC dendrites (See Supplementary Fig. 1a for a cross-sectional view). Starburst varicosities, which release ACh and GABA, are distributed over their distal dendrites (markers in Supplementary Fig. 1b). **b** Starburst dendrites respond preferentially to centrifugal motion (soma-to dendrite); their varicosities "wraparound" and make strong synaptic contacts selectively with DSGCs encoding the opposite direction (nDSGC; arrows indicate directional preferences). These are considered 'null connections' because stimulation of the starburst through a patch electrode evokes large GABAergic responses that dwarf the cholinergic responses co-activated in the synaptically connected nDSGCs. Stimulation of the same starburst evokes only cholinergic (but not GABAergic) currents in DSGCs encoding the opposite direction (pDSGC) through 'preferred' connections. The differential transmission of ACh/GABA aids in shaping direction selectivity[31,32], but its cellular basis remains unknown. **c** Flat-mount view of the reconstructed dendrites of two opposite coding DSGCs (blue/pDSGC, green/nDSGC) and a starburst (red) (the coding preferences of the DSGCs are shown by the black arrows; see Supplementary Fig. 1b for how the coding preferences of DSGCs were anatomically determined[25]). Blue arrow (top left) points to the location of the synapse between the red starburst and green nDSGC. Also shown are the dendrites of a Type 5o bipolar cell (pink), which makes a ribbon synapse with the red starburst and pDSGC. **d** A zoomed-in version of the synapse shown in (**c**). The starburst dendrite (red) makes a wraparound null contact (white spheres) with the nDSGC (green), and a peripheral contact with the pDSGC (blue). **e** Cross-sections illustrating the ultrastructural features of the tripartite complex in (**d**). The electron-dense postsynaptic density is evident at the wraparound synapse with nDSGC but not with the "peripheral" point of contact with the pDSGC. **f** Distribution of distances between a starburst varicosity and its closest peripheral contact ($n = 191$ varicosities contacting 6 independent ON-OFF DSGCs in 1 retina). Dotted line shows the cumulative distribution function. **g** The distribution of the directional preference of the DSGCs making peripheral contacts ($n = 115$ varicosities, 6 DSGCs from 1 retina), shown relative to the DSGCs receiving the wraparound contacts. The data is shown as a relative fraction of the total varicosities. Source data are provided as a Source Data file for Fig. 1f, g.

correlations in sEPSC activity as a ratio of the peak amplitudes of the STA and the average sEPSC. Over the population, the magnitude of correlated activity was $0.12 \pm 0.01$ ($n = 7$ pairs; Fig. 2d). The low rate of sEPSCs (~3 Hz) made it highly unlikely that the strength of the observed correlations arose by chance from independent sources (probability = $0.003^2 * 100 = 0.0009\%$). Thus, we interpret these correlated sEPSCs as direct

evidence that diffusion of ACh released from single vesicles can rapidly activate currents at multiple postsynaptic sites on different neurons. Such "multi-directed" cholinergic transmission challenges the idea that rapid sEPSCs signify transmission through conventional nanodomain-coupled synaptic mechanisms.

Several control experiments and analyses bolster these conclusions. The distribution of the peak amplitudes of the correlated

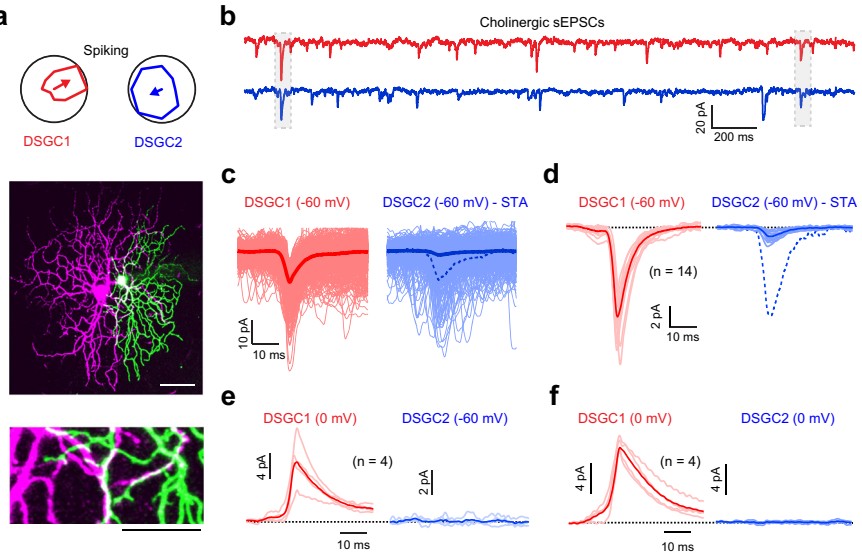

**Fig. 2 Rapid, multi-directed cholinergic transmission. a** Polar plots depicting the directions encoded by a pair of adjacent DSGCs, based on their spiking responses to spots of light moving in eight different directions (top). The middle panel shows the dendritic morphologies of the same pair of DSGCs imaged using two-photon microscopy (scale bar 50 μm). The bottom panel shows a magnified view where the dendrites of the two DSGCs co-fasciculate (scale bar 25 μm). **b** Simultaneous paired voltage-clamp recordings from the same DSGCs shown in (**a**), illustrating spontaneous cholinergic currents (sEPSCs; which are blocked by a broad spectrum nAChR antagonist 1 μM DHβE or by the α6*-specific nAChR blocking peptide PelA-5069; see Supplementary Fig. 2). A few of the sEPSCs are synchronized in the two DSGCs (boxed region). This indicated that ACh release from single vesicles can rapidly activate receptors at two postsynaptic sites, i.e., ACh is multi-directed. **c** Spontaneous events were detected in a reference DSGC (red) and aligned to their peak (1054 events). Currents over these same time periods in the nearby DSGC are shown in blue. Dark lines indicate the average current over all the sweeps (STA: the sEPSC triggered average). Normalizing the STA to the average sEPSC recorded in the reference cell reveals that its waveform is kinetically indistinguishable from that of the average sEPSC (dashed line). **d** The average sEPSC (red) and the average STA (blue) across 7 DSGC pairs from 5 retinas. Dark lines indicate the population average and the light lines illustrate responses in individual DSGCs. The dashed blue line shows the average STA normalized to the sEPSC. Note that for each recording two unique values are obtained by considering each DSGC as a reference cell. **e**, **f** Correlations are not observed between sIPSCs and sEPSCs (n = 4 pairs from 2 retinas) (**e**), or between sIPSCs (n = 4 pairs from 3 retinas) (**f**), recorded simultaneously in a subset of DSGCs shown in (**d**). Dark lines indicate the population average and the light lines illustrate responses in individual DSGCs. Source data are provided as a Source Data file for Fig. 2b–f.

sEPSCs was similar to the population distribution of sEPSCs (Supplementary Fig. 3; Supplementary Table 1), indicating that correlated events did not represent a select population of larger events that may be associated with multivesicular release[7,37]. Furthermore, spontaneous inhibitory currents (sIPSCs) and sEPSCs were not correlated (n = 4 pairs; Fig. 2e), arguing that even if the same varicosity contains both GABA and ACh vesicles, they must be packaged and released independently. In addition, sIPSCs were not correlated across neighboring DSGCs, confirming that starburst GABA transmission and other inhibitory inputs to DSGCs work through classic synapses (n = 4 pairs; Fig. 2f). Thus, it is likely that ACh released from single starburst varicosities spreads to co-activate nAChRs on at least two postsynaptic sites, residing on distinct neurons.

If tripartite connections mediate synchronized sEPSCs, then it might be expected that the extent of synchronization would be closely related to the degree to which the dendritic arbors of neighboring DSGCs overlap with each other. To test this idea, we first quantified the degree of dendritic overlap by measuring the Euclidean distances between each point in the DSGC dendritic tree (~1 μm segments) and the nearest dendritic point in the neighboring DSGC (nearest neighbor distances; NNDs; Fig. 3a–c; See "Methods")[38]. Plotting the NNDs in space allows for a direct visualization of the proximity of dendrites between two DSGCs. Such plots show few points of direct dendritic overlap (red), with much of the tree in close proximity to the adjacent DSGC (warm colors) (Fig. 3a, b). This is not surprising given that dendrites of DSGCs co-fasciculate within the 'honeycomb' starburst plexus

(Fig. 1a; Supplementary Fig 1a). The degree of dendritic overlap is also reflected in the cumulative distribution functions, where the NNDs for pairs of DSGCs with greater dendritic overlap tended to rise more sharply with distance compared to pairs with less overlap (Fig. 3c).

Having determined the extent of dendritic overlap, we next asked to what degree it predicts the strength of the sEPSC correlations. We found that the fraction of dendritic crossings (NND = 0 μm) provided an underestimate of the correlation strength between sEPSCs in a given pair of DSGCs (Fig. 3d; paired t test p value = 0.002). This suggested that ACh likely spreads and generates synchronized responses over a broader spatial scale. However, considering the fraction of dendrites with NNDs < 2 μm or <3 μm led to overestimates of the physiological correlations (Fig. 3f, g; paired t test p values = 0.002, 0.0001, respectively). When the fraction of dendrites with NNDs < 1 μm were considered, a good estimate of the sEPSC correlation strength was achieved, suggesting that ACh may activate receptors over this range (Fig. 3e; paired t test p value = 0.062). The close correspondence between the dendritic overlap over the < 1 μm scale and coincident sEPSCs indicates that most starburst inputs that can be shared, in fact produce synchronized postsynaptic responses, supporting our anatomical findings (Fig. 1).

**Multi-directed vs. spillover transmission.** Two lines of evidence distinguish multi-directed from conventional forms of "spillover" transmission as defined in many regions of the nervous

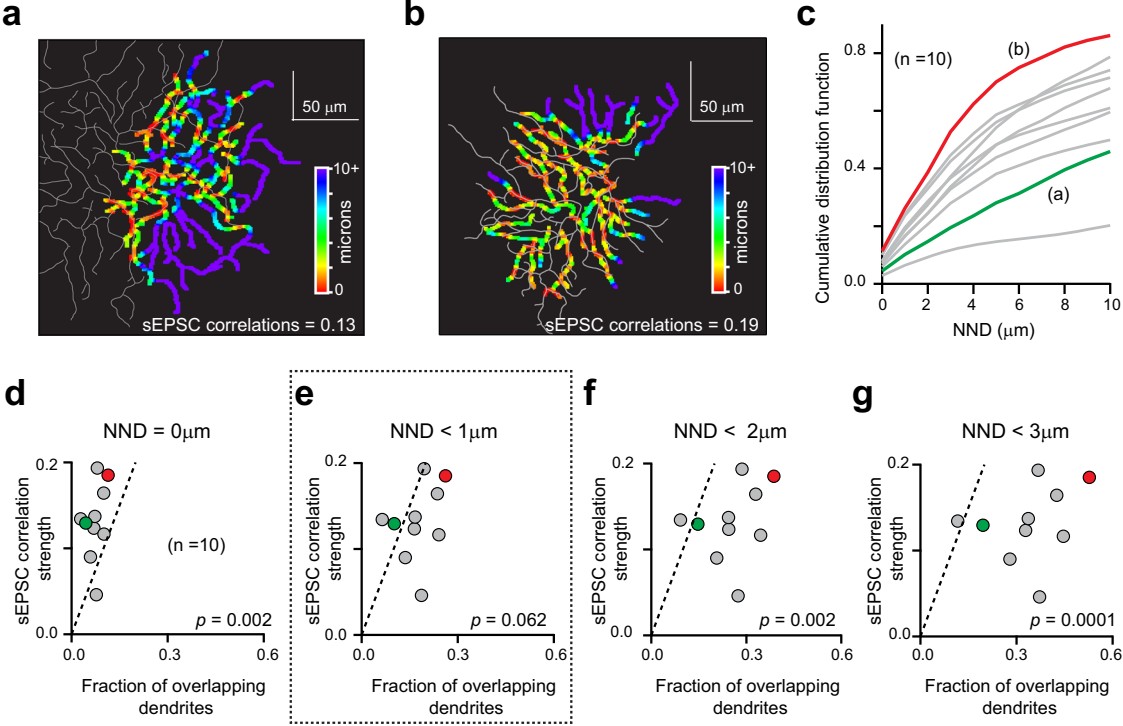

**Fig. 3 Structure/function correlations suggest that quantal ACh signals are mediated over a broad spatial scale (~1 μm). a** Reconstructions of the OFF dendritic arbors of two DSGCs. For illustrative purposes the dendritic arbor for one of the DSGC is thickened using a 3 × 3 average filter and color-coded to indicate the nearest neighboring dendrite distance (NND; see text). For simplicity, the maps of the ON arbors are not shown. NNDs were computed for 1 μm dendritic segments. The strength of the correlation between sEPSCs recorded in this pair of DSGCs is indicated at the bottom. The correlation strength was quantified as the ratio of the peak amplitudes of the STA and average sEPSC. **b** Similar to (**a**), but for a different pair of DSGCs with greater dendritic overlap and higher correlated activity. **c** Cumulative distribution functions of NNDs for all the segments in each DSGC ($n = 10$ DSGCs from 5 pairs). Green and red traces indicate the distributions for the DSGCs shown in (**a** and **b**), respectively. **d–g** Comparison of the correlation strength of the sEPSCs recorded in DSGC pairs to the fraction of overlapping dendrites (calculated from the NND distribution; see methods), for four different distances (0, 1, 2, and 3 μm; dotted line is the unity line). The colored data points indicate the values obtained from the DSGC pairs shown in (**a** and **b**). The overlap in dendrites is only predictive of the correlation strength when the fraction of dendrites within 1 μm is considered (**e**). This indicates that correlated ACh signals are mediated at dendritic crossings within ~ 1 μm of each other suggesting the tripartite complex (Fig. 1d, e) as the synaptic substrate. $p$ indicates the level of significance for paired $t$ tests. Source data are provided as a Source Data file for Fig. 3c–g.

system[4,5,7,8,39,40] (Fig. 4a, b). First, close inspection of individual pairs of correlated sEPSCs in nearby DSGCs revealed striking correlations in their decay kinetics, and to a lesser extent, in their rise-times and amplitudes (Fig. 4c, d). If cholinergic transmission to peripheral sites was mediated via spillover transmission, then we would expect sEPSCs derived from peripheral sites to be smaller in amplitude, and slower, compared to sEPSCs null synaptic connections[4,5,7]; and thus, during spontaneous activity, the synchronized events would be expected to be anti-correlated in their amplitude and kinetics, which we did not observe.

Second, when we examined unitary events evoked by stimulating single preferred- or null-side starbursts using a patch electrode, the postsynaptic cholinergic events had similar strengths and kinetic properties (Fig. 5). In these experiments, we replaced extracellular $Ca^{2+}$ with $Sr^{2+}$ to desynchronize vesicle release[41] and make quantal events easily discernable (Supplementary Fig. 4; Supplementary Table 1). In the presence of $Sr^{2+}$, brief depolarization of starbursts led to long-lasting barrages of asynchronous EPSCs (aEPSCs) (Fig. 5a). These aEPSCs had similar properties to sEPSCs (Fig. 5b, d, e; Supplementary Table 1), and thus could be used to characterize the properties of individual null and preferred connections. Notably, the amplitudes, frequency, and kinetics of the isolated aEPSCs were indistinguishable for both preferred- and null-starburst stimulation (Fig. 5c–e; Supplementary Table 1). These

findings directly argue against the idea that spillover is the main mode of transmission at preferred starburst connections.

In summary, based on the framework that has been meticulously laid out previously[2,4,5,8,12,39,40,42,43], our empirical results reveal that cholinergic transmission in the starburst-DSGC network resembles both synaptic and non-synaptic forms of transmission, and thus strictly conforms to neither mode. While several pre- and postsynaptic factors may help normalize responses at preferred and null connections (Supplementary Fig. 5; see Discussion), the precise mechanisms underlying multi-directed transmission–where rapid cholinergic signals are conveyed to two or more postsynaptic elements with equal efficacy–remain to be elucidated.

**ACh is locally, but not globally, tuned for direction.** So far, our functional measurements were made under resting conditions, and it was not clear whether cholinergic signals would remain confined when stimulated with light. How far ACh spreads will have a direct bearing on what kind of excitatory information the direction-selective dendrites of starbursts convey to downstream neurons. For instance, diffusion over long distances would result in accumulation of ACh from multiple starburst release sites (encoding different directions), and render signals non-directional. However, if ACh remains confined, then cholinergic input at individual DSGC dendrites might be expected to be

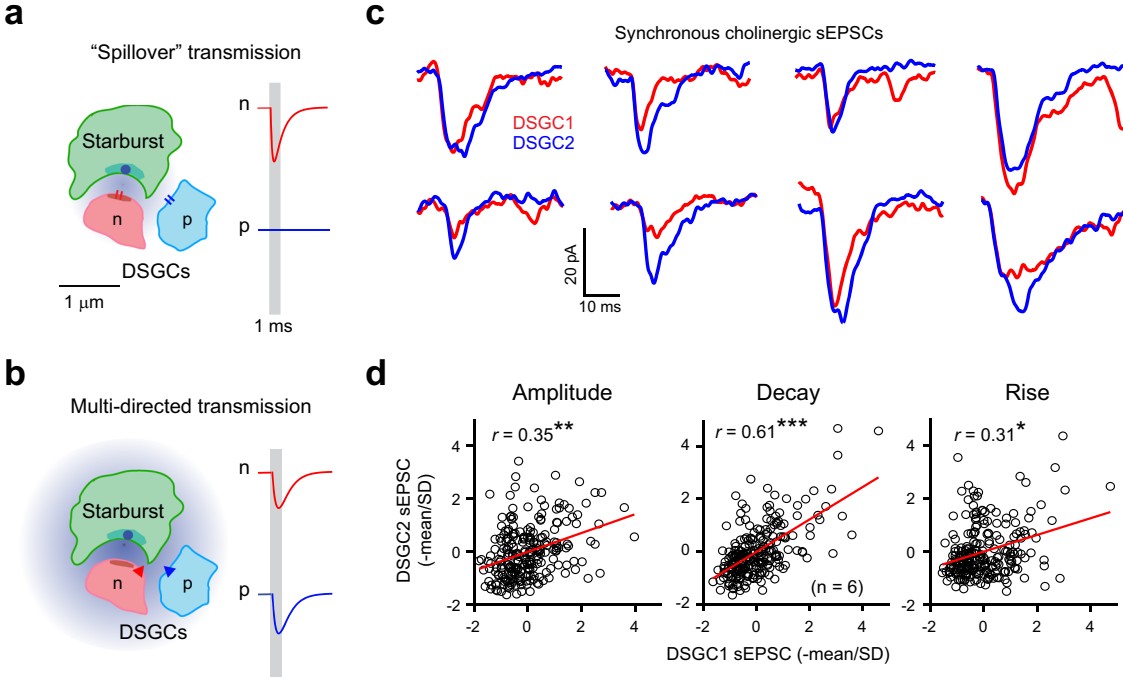

**Fig. 4 Synchronized sEPSCs are similar in amplitude and kinetics indicative of multi-directed transmission.** Cartoons of the tripartite complex (Fig. 1e) highlighting the differences between conventional "spillover" (**a**)[7,39,42,43] and multi-directed forms of ACh transmission (**b**). **a** ACh released from the starburst terminal rapidly activates low-affinity nAChRs mainly at null connections (n) giving rise to quantal EPSCs (red trace on right). A small fraction of ACh molecules also diffuse to preferred connections (p) where they mediate slower, weaker responses that generally fall below the detection limit of electrophysiological recordings (blue trace on the right). However, during more intense activity, ACh from multiple neighboring synapses can "spillover" and generate sizable responses at preferred sites, which could in theory explain the observed discrepancy between the anatomical and functional ACh connectivity[25,29,31]. **b** The central proposal here is that ACh signaling occurs over a broad spatial scale giving rise to "multi-directed" transmission. Unlike spillover transmission, ACh release from a single vesicle is equally efficacious at preferred and null sites (right), likely mediated by receptors with high binding affinity (triangle). A variety of factors including the geometry of the synaptic cleft, the concentrations of ACh packaged in single vesicles and the precise biophysical properties of nAChRs (e.g., low vs. high binding affinity) could dictate whether spillover or multi-directed transmission occurs (Supplementary Fig. 5). **c** Synchronous cholinergic sEPSCs observed in DSGC pairs indicate that ACh release from a single vesicle activates two sites with comparable efficacy, supporting the multi-directed transmission model (**b**). **d** The peak amplitudes (left), decay constants (middle) and rise times (right) of the synchronous events plotted against each other (241 events from 6 DSGC pairs, across 5 retinas). The red line indicates the line of best fit, and $r$ indicates the Pearson's correlation coefficient which is significantly higher than a shuffled distribution (*$p = 0.043$, **$p = 0.0055$, ***$p = 9.7 \times 10^{-14}$; two-tailed z-test; see methods). The data were standardized to the mean and standard deviation (SD) to avoid artefacts that could be introduced by cell-to-cell variability in these parameters. Source data are provided as a Source Data file for Fig. 4c, d.

directional, reflecting the directional tuning of the starburst release site. To assess ACh dynamics under more naturalistic stimulation conditions, we used a G-protein-coupled receptor activation-based sensor (ACh 3.0), which has an ACh binding affinity of ~2 μM, that is roughly comparable to that of the endogenous α6*-nAChRs[44,45].

To determine the spatial extent of ACh released from single starburst varicosities, we expressed ACh3.0 selectively in starbursts (using the ChAT-Cre line; see "Methods") and loaded single starbursts expressing the sensor with SeTau-647 (red indicator) through a patch electrode (Fig. 6a). Brief depolarizations of single starbursts evoked spatially localized fluorescence transients (space constant, $\lambda = 0.87 \pm 0.03$ μm; decay time constant, $\tau_{\text{decay}} = 141 \pm 11$ ms; $n = 81$ sites from 12 starbursts; Fig. 6b–d) that mapped to individual varicosities of the stimulated starburst, but were not necessarily centered on them (Fig. 6c, d; Supplementary Fig. 6). Responses could only be detected in $35 \pm 3\%$ of the imaged varicosities, suggesting non-uniform release properties of ACh across starburst dendrites (Fig. 6c; Supplementary Fig. 6).

During natural stimulation of the circuit with moving light bars, however, the contribution of individual varicosities to the ACh response was less evident, regardless of whether we expressed

ACh3.0 selectively in starburst or DSGC dendrites (a newly identified Oxtr-T2A-Cre line that labels predominantly nasally-tuned ON-OFF DSGCs was used to identify DSGC dendrites; Supplementary Fig. 7). Nevertheless, the correlations in the trial-to-trial variability of peak ACh3.0 responses decayed over relatively short distances ($\lambda \sim 1.5$ μm; Fig. 7b; see "Methods"), likely reflecting the contribution of ACh released from single starburst varicosities.

Notably, signals measured in small regions of interest in DSGC dendrites (ROIs, identified by mapping regions over which noise was correlated; see "Methods") were tuned for different directions (Fig. 7a, c, d, e; direction selectivity index [DSI] $= 0.35 \pm 0.01$; $n = 723$ ROIs; See Eq. (4)), indicating that ACh release from varicosities was directional, similar to the Ca$^{2+}$ responses observed in starburst varicosities using two-photon imaging[33,34]. In this experiment, ACh signals were only measured from dendrites of single ON-OFF DSGCs that were loaded with a red indicator (Alexa 594) through a patch-electrode prior to the imaging session. Perturbing the spread of ACh in the extracellular space by blocking ACh esterase (using 50 nM AMB) reduced the direction tuning at most sites (Fig. 7e; DSI $= 0.19 \pm 0.01$; $p = 3.03 \times 10^{-65}$, Wilcoxon signed-rank test). Thus, ACh esterase limits the spread of ACh to within ~1–2 μm of the release site,

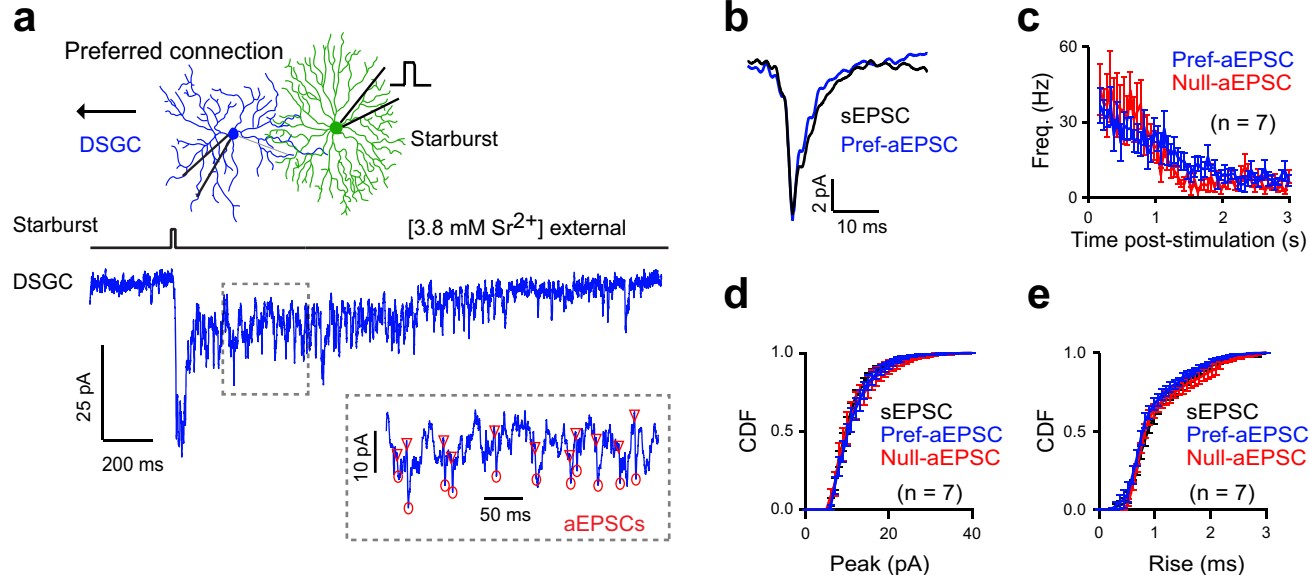

**Fig. 5 Quantal properties of preferred and null starburst cholinergic connections to DSGCs. a** Replacing extracellular $Ca^{2+}$ with $Sr^{2+}$ results in the desynchronization of vesicle release, providing insights into the unitary properties of preferred connections (see Supplementary Fig. 4 for responses elicited in $Ca^{2+}$ external). Inset shows miniature-like fast asynchronous events (aEPSCs) observed after brief depolarization of the starburst to activate preferred connections. Markers indicate the rising phase of detected aEPSCs. **b** The waveforms of the average aEPSCs measured in (**a**), overlaid with the average sEPSCs in the same cell. **c** The instantaneous frequencies (Freq.) of aEPSCs mediated by preferred (pref) or null connections ($n = 7$ pairs each). Data represented as mean ± SEM. Cumulative frequency distribution (CDF) of the peak (**d**) and rise times (**e**) of the sEPSCs compared to the aEPSCs, elicited by preferred- and null-connections ($n = 7$ pairs each from 9 retinas). Data represented as mean ± SEM. Source data are provided as a Source Data file for Fig. 5a–e.

and helps maintain the specificity of starburst cholinergic inputs during natural patterns of activity.

Color maps of the DSI and preferred angles show that the strength of the direction tuning was heterogeneous across the dendritic tree (Fig. 7a), and had no apparent relation with the DSGC's preferred-null axis (Fig. 7d; the preferred-null axis was determined from the DSGC's spiking responses; Supplementary Fig. 7n). Furthermore, the average peak amplitudes of the ACh signals across the dendritic tree were also non-directional (Supplementary Fig. 8). It is important to note that the organization of the presynaptic wiring reflected in the imaging experiments is not contingent on the precise spatiotemporal profiles of the ACh signals, which may be distorted to some extent by the slow kinetics of the ACh sensor (Fig. 6). The lack of global order in the strength of local cholinergic responses, along with the random distribution of the direction tuning, argues against the idea that cholinergic excitation is biased toward the DSGC's preferred direction[27,46].

## Discussion

**Multi-directed cholinergic transmission.** In theory, the symmetrical cholinergic receptive fields observed in DSGCs could arise from synaptic or non-synaptic mechanisms (Fig. 1b). Spillover or volume transmission is expected to make ACh insensitive to the asymmetric distribution of wraparound synapses (in contrast to GABA, which mediates asymmetric inhibition)[25,29,31]. Alternatively, based on the "phasic" nature of evoked cholinergic currents observed upon starburst stimulation[27–31], it has also been suggested that ACh is mediated by synaptic mechanisms[27]. In this case, symmetrical cholinergic excitation must be mediated by a small fraction of wraparound synapses (~20% of the total), such that the overall distribution remains asymmetrical[25]. At first, the robust expression of rapid spontaneous, asynchronous cholinergic EPSCs seemed to support the idea that ACh is mediated by

conventional point-to-point forms of synaptic transmission[27]. However, several lines of evidence argued against the idea that cholinergic excitation is mediated by a sparse synaptic input.

First, ACh sensor imaging revealed a high density of independently localized cholinergic inputs in DSGC dendrites (2-3 sites/10 μm; Fig. 7a), which is similar to the number of GABAergic synapses found anatomically on DSGCs[25]. Second, our optical imaging experiments also indicated that ~35% of starburst varicosities release ACh (Fig. 6; Supplementary Fig. 6), which is high compared to the ~20% expected for the synaptic model (described above). Third, spontaneous cholinergic events were found to occur at a ~3 fold higher frequency compared to spontaneous GABAergic IPSCs. This suggested a larger number of excitatory versus inhibitory functional connections[41], which is also inconsistent with the sparse synaptic ACh model.

A final piece of evidence that conclusively reconciles divergent views on the nature of cholinergic transmission comes from the observation that a significant fraction of the sEPSCs was synchronized in neighboring DSGCs. This central result directly demonstrates that ACh spreads beyond the confines of single wraparound synapses, and yet mediates rapid responses. Given that sEPSCs were synchronized on a fine temporal scale, it indicates that receptors in both cells are positioned close to the release site. This makes the tripartite complexes the most likely anatomical substrate for mediating ACh signals, notwithstanding the observation that peripheral contacts do not have any discernable postsynaptic ultrastructural specializations. Together, these considerations suggest that the symmetrical cholinergic receptive fields of DSGCs are generated by an alternate transmission mode, which bears functional and anatomical similarities to both synaptic and non-synaptic models, but does not conform to either. Given that this form of transmission rapidly activates multiple downstream targets, we refer to it as multi-directed transmission.

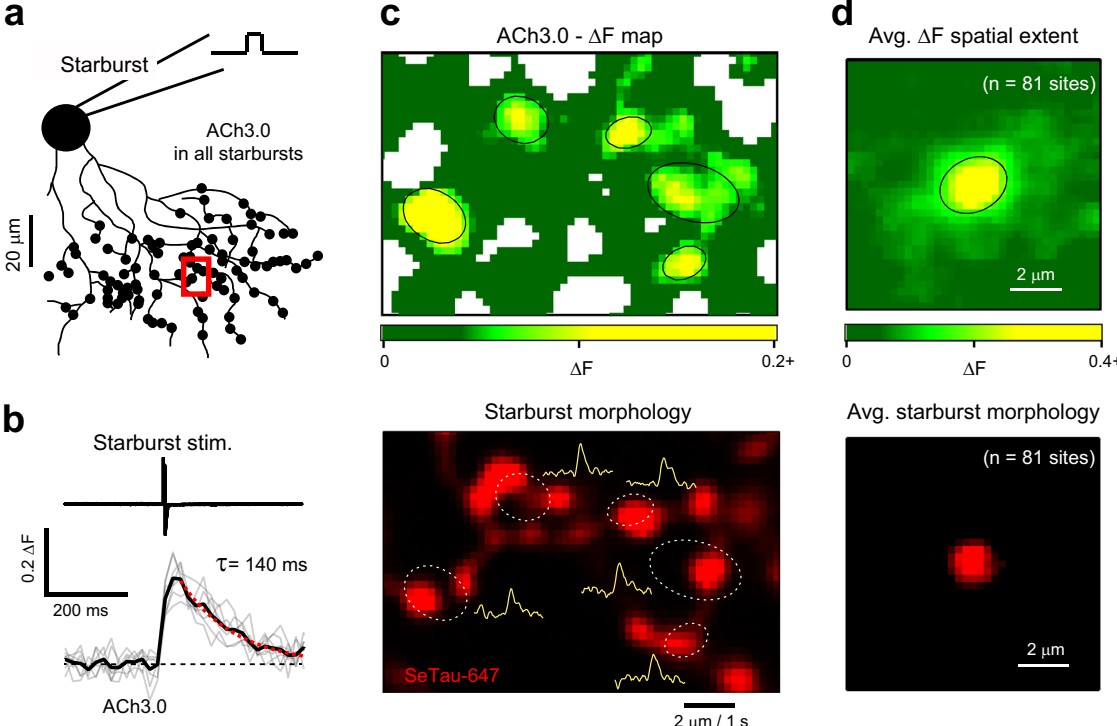

**Fig. 6 Two-photon imaging of ACh activity using a genetically encoded sensor reveals that ACh release is localized to within a micron of its release site. a** A genetically encoded ACh sensor (ACh3.0) was selectively expressed in many starbursts, one of which was stimulated with a patch electrode. A red indicator (SeTau-647) was included in the electrode solution, to reveal the starburst's morphology. In total, data was collected from 12 starbursts across 8 retinas. **b** The time course of changes in ACh3.0 sensor fluorescence ($\Delta F$; see Eq. 2) in the boxed region shown in (**a**), upon depolarizing the starburst (top trace; Starburst stim.). The gray traces show individual trials while the black trace indicates the average over 7 trials. The red dotted line represents an exponential fit to the response decay (decay constant ($\tau$) = 140 ms). **c** A high-resolution spatial map of the peak ACh3.0 responses in the boxed region shown in (**a**) (top; average over 7 trials). The white space indicates regions without ACh3.0 expression. The black contours indicate the 1 standard deviation level of a 2D Gaussian fit of the ACh3.0 responses. The bottom image shows the morphology of the stimulated starburst mapped on to the ACh3.0 response sites (dashed white contours). The yellow traces indicate the response timecourse (filtered) at each site. One such window was imaged from each starburst independently. **d** The average (avg.) spatial profile of the ACh3.0 responses (top) and the starburst morphology (bottom) for 81 sites measured in 12 starbursts (from 8 retinas). This indicated that the ACh3.0 responses were localized to within a micron of the starburst varicosities. Source data are provided as a Source Data file for Fig. 6b.

**Biophysical mechanisms underlying multi-directed transmission**. Multi-directed transmission is similar, yet distinct from "engineered spillover" mechanisms at photoreceptor ribbon synapses that supply glutamate to multiple postsynaptic elements[47]. At these synapses, glutamate receptors expressed by different postsynaptic cells are positioned at specific distances away from the release site, ranging from 20 nm to several hundred nanometers, each tuned to the local concentration profile of glutamate. Consequently, the amplitude and the kinetics of the response change systematically amongst postsynaptic neurons. Even cells that have their receptors positioned ~1 μm away from the release site exhibit synaptic-like responses, rising in less than a millisecond[7]. However, the important difference between cholinergic transmission and engineered spillover is that the cholinergic EPSCs at preferred and null connections appear to be similar in amplitude and kinetics despite the apparent difference in distances from the release site, indicating that distal sites are exposed to levels of ACh that are enough to generate rapid responses. Thus, rapid cholinergic signals appear to be transmitted over a relatively broad spatial scale, unlike conventional synapses or even photoreceptor ribbon-synapses.

Several factors may contribute to the relatively high concentrations of ACh experienced at distal sites (Supplementary Fig. 5). For instance, the geometry of the extracellular space is known to

directly impact how far neurotransmitters spread from their points of release. If ACh spread was constrained within a 2D space between the pre- and postsynaptic elements, then, as previously noted, the spatial gradient of ACh would be less prominent than if the extracellular space was more porous, allowing ACh to spread in 3D[48] (Supplementary Fig. 5a). In addition, the vesicular concentration of ACh might be high compared to GABA and glutamate[35,42], which could ensure that sufficient levels are reached at distal preferred sites. It is also possible that the biophysical properties of the nAChRs enable fast gating at relatively low concentrations of agonist. Indeed, a simple kinetic model of the tripartite synapse in which nAChRs were placed 0 μm or ~1 μm from the release site (to mimic the null and preferred connections, respectively) could produce similar responses, provided their agonist binding affinity was high (similar to that reported for α7-nAChRs; Supplementary Fig. 5b–d). Lower desensitization rates also enabled distal receptors to temporally integrate ACh signals, which directly aided in the equalization of responses at proximal and distal sites (Supplementary Fig. 5e). Estimating the synaptic cleft volume without distorting the extracellular space during fixation[49], and determining the biophysical properties of α6-AChRs along with their precise subcellular distributions, are important steps that are needed to confirm the details of the synaptic mechanisms governing multi-directed transmission.

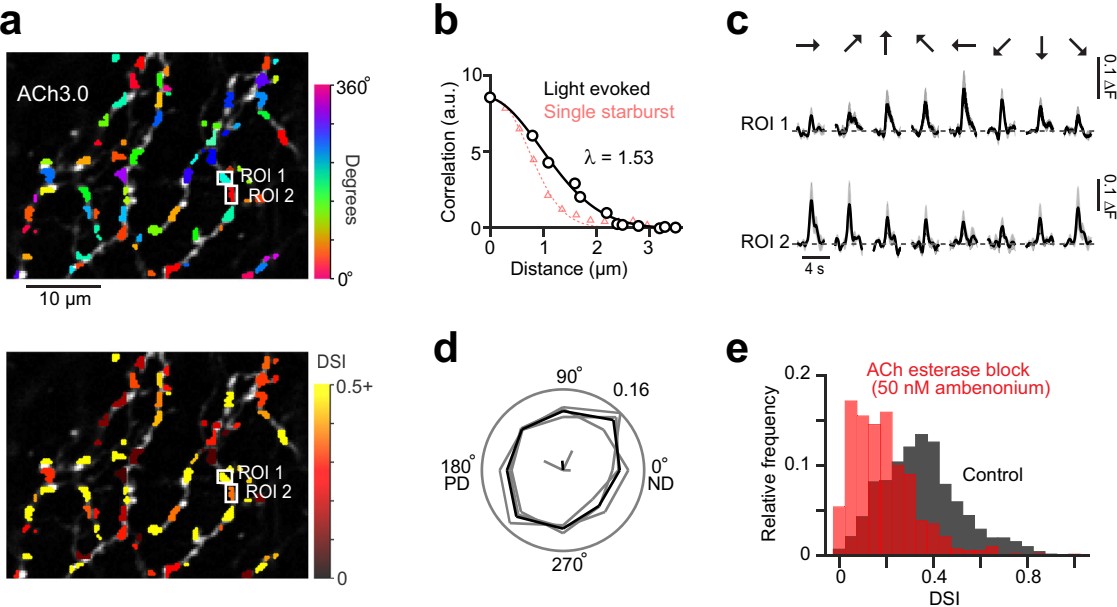

**Fig. 7 ACh signals in DSGCs are locally, but not globally, tuned for direction. a** Two photon image of DSGC dendrites selectively labeled with ACh3.0, using the Oxtr-T2A-Cre mouse line (see Supplementary Fig. 7 for a more detailed characterization of this new mouse line). The colors indicate the preferred direction for each site (top), and its tuning strength (bottom), quantified from the responses to spots of light moving in 8 directions (See methods). Tuning strength was quantified from the direction selectivity index (DSI; See Eq. 4), where 0 indicates non-tuned, and 1 indicates responses only in the preferred direction. In total, 10 such field of views were imaged from 3 DSGCs. **b** Noise correlations (trial-to-trial fluctuations around the mean ACh3.0 signal) plotted over distance ($\lambda$, space constant; black). The spatial decay of ACh signals associated with single starburst stimulation measured in Fig. 6d is shown for comparison (red) (a.u., auxiliary units). **c** Example ACh ΔF signals evoked by spots moving in 8 different directions, measured in the two ROIs shown in **a**. Data represented as mean (black) ± SEM (gray) from 4 trials. **d** Relative frequency histogram of the preferred directions of all the ROIs in polar form ($p = 0.81$, $n = 723$ ROIS, 3 DSGCs; Hodges-Ajne non-parametric test for angular uniformity of the data) indicates a random distribution of angles with respect to the DSGCs preferred and null directions (PD and ND, respectively). **e** Relative frequency histogram of the DSIs of all the ROIs in control, and in the presence of an ACh esterase blocker (50 nM ambenonium). Source data are provided as a Source Data file for Fig. 7b, d, e.

**Functional implications for direction selectivity**. Previous electrophysiological analyses indicated that under different conditions, cholinergic signals mediated by preferred connections appear to be stronger[27,46], weaker[28], or equal in magnitude[31,32,50] relative to those mediated by null connections. These results have generally been difficult to interpret due to space-clamp considerations[51]. Here, our ACh imaging indicates that cholinergic excitation is symmetrical on a global scale. However, it should be noted that these experiments only tested select ON-OFF DSGC types, and it is possible that other ON-OFF and ON DSGC types might have biased distributions of preferred cholinergic synapses.

At first glance, the lack of global order in the cholinergic excitation appears consistent with existing models in which direction selectivity arises from the integration of non-directional excitation and directionally tuned inhibition[25,30,31,34,50,52–54]. But, several lines of evidence suggest that nonlinear integration of synaptic inputs occurs over subregions of DSGC dendrites[29,53] that can be as small as ~10 μm in length[52]. In this case, directional cholinergic input would shape local dendritic responses. When cholinergic inputs align with the null direction (i.e., the wraparound connections), the co-transmission of GABA occurring at that same site would effectively cancel excitation. Fine-scale coordination of excitation and inhibition is conditional on ACh signals being highly compartmentalized and directional. For cholinergic inputs aligned with the preferred direction, the chance that they receive coincident GABA input is low. In this way, the net cholinergic drive emerges strongly directional. Thus, locally tuned cholinergic input suggests a new role for ACh in the direction-selective dendritic computation.

**Rapid cholinergic signals are transmitted over a broad spatial scale**. Rapid cholinergic responses have been observed in many regions of the brain[13–19], challenging the conventional notion that ACh signals are solely mediated via spillover or volume transmission[20–24]. However, to date, electron microscopy studies have not provided a consensus regarding the extent to which cholinergic terminals form well-defined synaptic structures[10,11,22,25,35]. What complicates these results is the recent realization that many cholinergic neurons—like the starburst—also release a second fast neurotransmitter (GABA[35,55–57] or glutamate[23,58]), often from the same terminal varicosities[23,35,59], making it difficult to determine the ultra-structural elements that are associated with cholinergic transmission. In this study, we took advantage of the known connectivity patterns in the retinal direction-selective circuit and compared the functional and anatomical properties of individual ACh versus mixed ACh/GABA connections. Based on the anatomy alone, the lack of ultrastructural specializations at the peripheral contacts pointed to volume or spillover transmission. In contrast, the physiology demonstrated the quantal nature of unitary cholinergic connections (made by both preferred and null starbursts), which is generally taken as clear evidence for conventional synaptic mechanisms[2,4,5,8,12,39,40,42,43]. These diametrically opposing views are easily reconciled once it is realized that rapid cholinergic signals are transmitted over a broad spatial scale. In this context, the debate regarding the nature of cholinergic transmission, which has engaged neuroscientists for decades, may be fundamentally flawed in assuming that cholinergic neurons utilize similar mechanisms to other fast neurotransmitters or slow neuromodulators.

Multi-directed transmission is an efficient way in which small populations of cholinergic neurons can powerfully modulate

network activity with temporal precision, through their sparsely distributed connections[13,15–18,60,61]. The idea that rapid ACh transmission occurs on a broad spatial scale compared to other fast neurotransmitter systems has also been suggested at cholinergic synapses in the chick autonomic systems[62] (but see Coggan et al.[42]), and in other regions of the brain[13,35,55], but confirming whether the transmission is multi-directed in these areas remains a challenge for future investigations. The conceptual and technical approaches presented here set the stage for testing this hypothesis.

## Methods

**Animals**. Electrophysiological experiments were performed using adult Hb9-EGFP (RRID: MGI_109160) or Trhr-EGFP (RRID: MMRRC_030036-UCD) crossed with ChAT-IRES-Cre (RRID: MGI_5475195) crossed with Ai9 (RRID: MGI_3809523). Acetylcholine imaging experiments were performed using Oxtr-T2A-Cre (strain: Cg-Oxtr$^{tm1.1(cre)Hze}$/J, Jackson laboratory stock: 031303) and ChAT-IRES-Cre mice under C57BL/6 J background. Oxtr-T2A-Cre x Thy1-STOP-EYFP (strain: Cg-Tg (Thy1-EYFP)15Jrs/J, Jackson laboratory stock: 005630) mice were used for characterization of the labeled cells (Supplementary Fig. 7). All mice were between two and sixteen months old in either sex. All procedures were performed in accordance with the Canadian Council on Animal Care and approved by the University of Victoria's Animal Care Committee, or in accordance to Danish standard ethical guidelines and were approved by the Danish National Animal Experiment Committee (Permission No. 2015-15-0201-00541; 2020-15-0201-00452).

**Physiological recordings**. Mice were dark-adapted for ~30–60 min before being briefly anaesthetized and decapitated. The retina was isolated in Ringer's solution under infrared light. The isolated retina was then mounted on a 0.22 mm membrane filter (Millipore) with a pre-cut window to allow light to reach the retina, enabling the preparation to be viewed with infrared light using a Spot RT3 CCD camera (Diagnostic Instruments) attached to an upright Olympus BX51 WI fluorescent microscope outfitted with a 40× water-immersion lens (Olympus Canada). The isolated retina was then perfused with warmed Ringer's solution (35–37 °C) containing 110 mM NaCl, 2.5 mM KCl, 1 mM CaCl$_2$, 1.6 mM MgCl$_2$, 10 mM dextrose and 22 mM NaHCO$_3$ that was bubbled with carbogen (95% O$_2$:5% CO$_2$). Perfusion rates were maintained at ~3 ml/min.

DSGCs were identified by their genetic labeling or by their characteristic direction-selective responses. Spike recordings were made with the loose cell-attached patch-clamp technique using 5–10 MΩ electrodes filled with Ringer's solution. Voltage-clamp whole-cell recordings were made using 4–7 MΩ electrodes containing 112.5 mM CH$_3$CsO$_3$S, 7.75 mM CsCl, 1 mM MgSO$_4$, 10 mM EGTA, 10 mM HEPES and 5 mM QX-314-Cl. The pH was adjusted to 7.4 with CsOH. For stimulating neurotransmitter release, starbursts were voltage-clamped with electrodes containing 112. 5 mM CH3CsO$_3$S, 7.75 mM CsCl, 1 mM MgSO$_4$, 0.1 mM EGTA, 10 mM HEPES, 10 mM AChCl, 4 mM ATP, 0.5 mM GTP, 1 mM ascorbate and 10 mM phosphocreatine (pH = 7.4). In some paired recordings, 100 μM Alexa-488 or 10 μM SeTau-647 (SETA Biochemicals, #K9-4150) was added to the internal solution to visualize the dendritic morphologies. The reversal potential for chloride was calculated to be −56 mV. The liquid-junction potential was calculated to be −8 mV and was not corrected. Recordings were made with a MultiClamp 700B amplifier (Molecular Devices). Signals were digitized at 10 kHz (PCI-6036E acquisition board, National 9 Instruments) and acquired using custom software written in LabVIEW. Unless otherwise noted, all reagents were purchased from Sigma-Aldrich Canada. Ambenonium chloride and UBP310 were purchased from ABCAM Biochemicals. DL-AP4, SR-95531 and CNQX were purchased from Tocris Bioscience.

**Virus injections**. The plasmid pssAAV-2-hSyn1-chI-dlox-Igk_(rev)-dlox-WPRE-SV40p(A) was generated by Viral Vector Facility at University of Zurich based on pAAV-hSyn-G that has been deposited to Addgene[45] (#121922;). The single-stranded AAV vector ssAAV-9/2-hSyn1-chI-dlox-Igk_(rev)-dlox-WPRE-SV40p (A) (1.2 × 10$^{13}$ vg/ml) was produced using the plasmid by Viral Vector Facility at University of Zurich. For intravitreal viral injections of the AAV, mice were anesthetized with an i.p. injection of fentanyl (0.05 mg/kg body weight; Actavi), midazolam (5.0 mg/kg body weight; Dormicum, Roche) and medetomidine (0.5 mg/kg body weight; Domitor, Orion) mixture dissolved in saline. We made a small hole at the border between the sclera and the cornea with a 30-gauge needle. Next, we loaded the AAV into a pulled borosilicate glass micropipette (30 μm tip diameter), and 2 μl was pressure-injected through the hole into the vitreous of the left eye using a Picospritzer III (Parker). Mice were returned to their home cage after anesthesia was antagonized by an i.p. injection of a flumazenil (0.5 mg/kg body weight; Anexate, Roche) and atipamezole (2.5 mg/kg body weight; Antisedan, Orion Pharma) mixture dissolved in saline and, after recovery, were placed on a heating pad for 1 h.

**Two-photon ACh sensor imaging**. Three-to-four weeks after virus injection into the eyes of Oxtr-T2A-Cre or ChAT-Cre mice, we performed two-photon imaging of ACh3.0 fluorescence signals on dendrites of the ON-OFF DSGCs, or starbursts. Only dendrites in the ON layer were imaged. Retinas were prepared following a procedure that was similar to that used for the electrophysiological recordings. The isolated retina was then placed under the microscope (SliceScope, Scientifica) equipped with a galvo-galvo scanning mirror system, a mode-locked Ti: Sapphire laser tuned to 940 nm (MaiTai DeepSee, Spectra-Physics), and an Olympus 60× (1.0 NA) objective. The ACh3.0 signals emitted were passed through a set of optical filters (ET525/50 m, Chroma; lp GG495, Schott) and collected with a GaAsP detector. Images were acquired at 8–12 Hz using custom software. The size of the imaging window was 40–60 μm in length and 80–120 μm in width (0.45 μm/pixel). Temporal information about scan timings was recorded by TTL signals generated at the end of each scan, and the scan timing and visual-stimulus timing were subsequently aligned during off-line analysis. In experiments where responses were evoked via depolarizing starbursts, ACh responses were brief, and hence, were measured at higher temporal resolution[52] (~50 Hz) (Fig. 6b). In these experiments, the size of the imaging window was typically 30 μm × 20 μm (0.28 μm/pixel; dwell time =1 μs/pixel).

**Imaging data analysis**. For analyzing the direction-selective properties of the ACh3.0 signals, regions of interest (ROIs) were determined by customized programs in MATLAB. First, the stack of acquired images was filtered with a Gaussian filter (3 × 3 pixels), and then each image was down-sampled by a factor of 0.8 relative to the original image using a MATLAB imresize function. The signals in each pixel were smoothed temporally by a moving average filter with a window size of 2 time-bin and resampled using the MATLAB interp function. To evaluate the spatial scales of the light-evoked acetylcholine signals, we calculated the noise correlation among the signals in pixels during static flash (300-μm diameter, 50% contrast), and plotted the correlation coefficient against the distance between pixels (Fig. 7b). The correlation was normalized as a score (C):

$$C_{p.q} = (c_{\tau=0} − M_c)/\mathrm{SD}_c \qquad (1)$$

where $c_{\tau=0}$ is the amplitude at time delay $\tau = 0$ in cross-correlation function between pixel $p$ and $q$, $M_c$ and SD$_c$ are mean and SD of the cross correlation function, respectively. The spatial scale of ACh responses were estimated by fitting the decay of noise correlations to a 2-D Gaussian function in MATLAB (Fig. 7b). We set a threshold of the correlation score, >2.5, to determine which pixels were to be included as a single ROI. The response of each ROI ($\Delta F(t)$) was calculated as

$$\Delta F(t) = (F(t) − F_0)/F_0 \qquad (2)$$

where $F(t)$ is the fluorescent signal in arbitrary units, and $F_0$ is the baseline fluorescence measured as the average fluorescence in a 1-s window before the presentation of the stimulus. After the processing, responsive pixels were detected based on a response quality index (RI):

$$RI^i = \left(R^i − \left(M_0^i + 2\mathrm{SD}_0^i\right)\right)/\left(R^i + \left(M_0^i + 2\mathrm{SD}_0^i\right)\right) \qquad (3)$$

where $R^i$ is a peak response amplitude during motion stimulus to direction $i$, and $M_0^i$ and SD$_0^i$ are mean and standard deviation of ACh signals before stimulus (1 s period), respectively. The ROIs with RI higher than 0.6 were determined as responsive (Supplementary Fig. 8a).

To evaluate the directional tuning, we calculated direction selectivity index (DSI):

$$\mathrm{DSI} = (\mathrm{R}_{Pref} − \mathrm{R}_{Null})/(\mathrm{R}_{Pref} + \mathrm{R}_{Null}) \qquad (4)$$

where $\mathrm{R}_{Pref}$ and $\mathrm{R}_{Null}$ denote the amplitude of ACh3.0 signals to preferred and null direction, respectively. DSI ranged from 0 to 1, with 0 indicating a perfectly symmetrical response, and 1 indicating a response only in the preferred direction. The preferred direction for individual ROIs was defined as an angle ($\theta$) calculated by the vector sum:

$$\theta = \tan^{-1}\left(\sum_i \sin i * R_i / \sum_i \cos i * R_i\right) \qquad (5)$$

where $i$ denotes the motion direction, and $R_i$ denotes the response amplitude.

**Light stimuli**. Visual stimulation was generated via custom software (Python and LabVIEW) written by Zoltan Raics. The generated stimulus was projected using a DLP projector (LightCrafter Fiber E4500 MKII, EKB Technologies) coupled via a liquid-light guide to an LED source (4-Wavelength High-Power LED Source, Thorlabs) with a 400-nm LED (LZ4-00UA00, LED Engin) through a band-pass optical filter (ET405/40x, Chroma). The stimulus was focused on the photoreceptor layer of the mounted retina through a condenser (WI-DICD, Olympus). The stimuli were exclusively presented during the fly-back period of the horizontal scanning mirror[54]. To measure the directional tuning, we used a spot (300-μm diameter, 2-s duration) moving in 8 directions (at 800–1200 μm/s) with a 50% contrast.

**Electron-microscopy analysis**. For ultrastructure, we used a previously published SBEM[26] data set (retina k0563). Voxel dimensions were 12 × 12 × 25 nm$^3$ (x, y, and

z, respectively). Potential ON-OFF DSGCs were first identified as ganglion cells with bistratified morphology in the IPL (Supplementary Fig. 1a). Next, wraparound synapses from starbursts on the dendrites of these cells were identified by their electron-dense membrane thickening laden with vesicles (Fig. 1e). Reconstruction of the starbursts confirmed their co-stratification with the ganglion-cell dendrites identifying it as a DSGC (Fig. 1c). Next, to estimate the closest peripheral contact, we retraced the dendrites of all the identified DSGCs in the volume (Fig. 1a), and examined the closest contact made by these DSGC dendrites to starburst varicosities outside the wraparound synapse. In this analysis, the preferred direction of the DSGC was estimated from the orientation of the synapse forming starburst dendrites (Supplementary Fig. 1b). All analyses were performed by tracing skeletons and annotating synapses using the Knossos software package (www. knossostool.org). Volumetric reconstructions of synapses were performed using ITK-SNAP (www.itksnap.org).

**Spontaneous EPSC analysis.** Asynchronous EPSCs (aEPSCs) were acquired from DSGCs for 1-s post-starburst stimulation, whereas spontaneous EPSCs in the same DSGC were acquired for 20 s after the aEPSCs (over multiple trials). Preferred- and null-starburst-DSGC connections were identified by estimating the preferred direction of the DSGC using extracellular spike recordings before performing paired recordings. In cases where only sEPSCs were acquired, DSGCs were recorded in a dark background for a period of 5–15 mins. Once acquired, the traces were low-pass filtered at 300 Hz, and fast-rising events were detected automatically using tarotools (sites.google.com/site/ tarotoolsregister/) in IgorPro. The events were detected using amplitude thresholds set between 5 and 7 pA. Multiple overlapping events observed during the asynchronous activity were separated using the default (2.5-ms) decay-constant parameter[41]. In the case of the experiments examining aEPSCs, peak amplitude and 20–80% rise times were calculated with in-built tarotools functions. In the case of experiments examining only sEPSCs, the events were manually checked before calculating the parameters using a custom-written script in IgorPro. During paired DSGC recordings, after detecting sEPSCs in a reference cell, the currents recorded in the second DSGC were averaged over the same time periods to compute the sEPSC triggered average (STA; Fig. 2c, d). The peak amplitude of the STA relative to the sEPSC yielded an estimate of the correlation strength of excitatory input to each DSGC. For the analysis of individual correlated events (Fig. 4c, d), only events that had peak amplitude above 5 pA in both DSGCs were selected.

**Nearest-neighbor analysis.** After estimating the sEPSC correlations, the dendritic morphology of the DSGCs was visualized with Alexa-488 and SeTau-647 using two-photon imaging techniques. The DSGC's dendritic arbor was traced using a custom-written script in IGORPro. ON and OFF arbors were analyzed separately. Note that the tracing algorithm did not preserve the thickness of the dendrites, and the thickness was artificially set to 1 μm. These traced skeletons were divided into 1 μm segments, and for each segment, the distance of the nearest segment in the neighboring DSGC was calculated (Fig. 3a, b). The nearest-neighbor distances for all the segments in a DSGC were binned (at 1 μm), and a cumulative distribution function was constructed (Fig. 3c). At this point, the distribution functions for ON and OFF arbors were averaged. The value of the cumulative distribution function (at a given distance) was considered as the fraction of overlapping dendrites (Fig. 3d–g).

**Statistical testing.** Sample sizes were not predetermined, and are comparable to contemporary electrophysiological and optical imaging studies[31,32,34]. Population data have been expressed as mean ± SEM and are indicated in the figure legends along with the number of samples. Student's t-test was used to compare values under different conditions, and the differences were considered significant if $p \leq 0.05$. For synchronized sEPSC properties (Fig. 4d), the significance of the observed correlation was determined by comparing the correlation coefficients of the original distribution with a shuffled distribution (i.e. the data from one cell in the pair was offset by one point). First, the correlation coefficients ($r$) of both the original and shuffled distributions were transformed to a z-score with a Fisher z-transformation,

$$z = 0.5 * (\ln(1 + r) - \ln(1 - r)) \tag{6}$$

Then a z-test statistic ($z_{observed}$) was computed using the formula

$$z_{observed} = \frac{z_{orig} - z_{shuff}}{\sqrt{\left(\frac{1}{n_{orig} - 3}\right) + \left(\frac{1}{n_{shuff} - 3}\right)}} \tag{7}$$

where $z_{orig}$ and $z_{shuff}$ are the z-scores of the original and shuffled distributions, and $n_{orig}$ and $n_{shuff}$ are the corresponding sample sizes. The two distributions were considered significantly different when $z_{observed}$ was greater than 1.96, the critical value for a significance level of 0.05 for a two-tailed z-test (adjustments were not made for multiple comparisons). For the imaging experiments, we used Wilcoxon signed-rank test to evaluate the effects of ambenonium (Fig. 7e). To evaluate the uniformity in the angular distribution, we used Hodges-Ajne test (Fig. 7d).

**Reporting summary.** Further information on research design is available in the Nature Research Reporting Summary linked to this article.

## Data availability

Source data are provided with this paper. The SBEM dataset, ek0563, is available at https://knossos.app/.

## Code availability

The custom codes used for sEPSC and NND analysis have been deposited in GitHub[63]. The codes for data acquisition are available upon request.

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

## Acknowledgements

We thank Dr. Kevin Briggman for making previous SBEM datasets available for further analysis. Dr. Kerry Delaney for his critical feedback on the manuscript. Tracey Michaels for performing AAV injections and help with mouse colony management; Zoltan Raics for developing our visual stimulation system, and Bjarke Thomsen and Misugi Yonehara for their technical assistance. Dr. Marla Feller for nGFP mice. Dr. Jamie Boyd for his help with IGOR software for 2 P imaging. This work was supported by grants awarded to A.M. (VELUX FONDEN Postdoctoral Ophthalmology Research Fellowship:27786); C.G (R01-NS042169); J.M.M. (NIH GM136430 and GM103801); Y.L.L. (General Research Program of National Natural Science Foundation of China (project 31671118), the NIH BRAIN Initiative (grant U01NS103558), the Beijing Brian Initiative of Beijing Municipal Science & Technology Commission (Z181100001518004), the Junior Thousand Talent Program of China, and by grants from the Peking-Tsinghua Center for Life Sciences and the State Key Laboratory of Membrane Biology at Peking University School of Life Science); D.B. (RO1 EY012793-19); K.Y. (Lundbeck Foundation: DANDRITE-R248-2016-2518; R252-2017-1060, Novo Nordisk Foundation, NNF15OC0017252, Carlsberg Foundation, CF17-0085, and European Research Council Starting, 638730) and G.B.A. (CIHR 159444).

## Author contributions

S.S. conducted all the electrophysiological and starburst imaging experiments shown in Figs. 2–6. A.M. performed the ACh imaging experiments shown in Fig. 7. Y.L. & M.J. developed ACh sensor; J.M.M. developed the nAChR antagonist; D.B conducted all the SBEB analysis in Fig. 1; K.Y. designed AAVs constructs & ACh imaging experiments; G.D. and C.G. constructed the kinetic nAChR model; B.M.B. wrote new software for image analysis. G.B.A and SS conceived the experiments and wrote the original manuscript.

## Competing interests

The authors declare no competing interests.
