## [Peer Review File · Nature Communications]

Reviewer #1 (Remarks to the Author):

The authors set out to determine the mode of cholinergic synapses between starburst amacrine cells and direction selective retinal ganglion cells. They elegantly combine several methods, including electrophysiology, anatomical reconstruction, and 2-photon imaging of ACh in order to compare different models of direct and diffuse synaptic transmission. The mechanism of cholinergic transmission and its role in establishing direction selectivity in the retina are active questions in the field, and this study is timely and relevant. I support publication of this paper with some revisions to improve the clarity of the manuscript.

My primary concern was that I found the manuscript a bit difficult to read which I think this stems from the fact that there are many different techniques, and most readers will not have equal experience with, or understanding of, each of them. I think the manuscript would be greatly improved by putting just a bit more explanation and unpacking into the descriptions of the techniques and key analyses.

A few examples:

- Figure 1 c-f. These schematics are extremely important and valuable. However, a bit more explanation would help. What exactly is the key difference between “spillover” (d) and “multi-directed” (f)? I assume it’s related to the position of the receptor in the schematic, but I don’t quite understand why that yields the different traces shown.
- The frequency of correlated sEPSCs (p. 5) – the authors state this is more than expected given the spontaneous firing rates of the cells. This seems right to me, but I’d like this logic unpacked, and assumptions either demonstrated or at least stated.
- The fact that correlated decay kinetics are indicative of ACh concentration profiles rather than deactivation (on p. 6) – why is this? Please unpack the logic for this.
- P7 “the fraction of dendrites of a given DSGC, that were within a certain distance of the dendrites of a second DSGC, grew rapidly as this distance increased”. This sentence was very confusing to me and a more careful explanation would help a lot. I would also ask for an intuition for a null hypothesis. The assertion that it “grew rapidly” suggests that there’s an assumption that it should either not grow or grow slowly? (perhaps minor detail, but I also don’t think “rapidly” is the correct word here as it implies a temporal component rather than spatial)
- Figure 3c: this appears to be a crucial analysis that ties together the different pieces. It is not clear to me what the p-values inset in the panels are nor how this supports the conclusion regarding the strength of correlation and distance. My understanding is that the correlation values (not shown) are too high at 0 um, and too low at 2 and 3 um, but close at 1um? Again, I think this can be resolved through just a bit more unpacking in the text.

A few of the supplementary figures seemed like they should be incorporated into the primary figures as they provided key evidence for the questions being addressed. Sup Figures 2 and 4 in particular.

Is there a citation for the fact that Oxtr-T2A-Cre labels nasal on-off DS-RGCs? Or is this a new detail regarding this Cre line that has not been shown – in which case please add a Supplemental Figure that documents this more. Eg. how selective is its expression among RGC types?

Reviewer #2 (Remarks to the Author):

Rapid 'multi-directed' cholinergic transmission in the central nervous system

This very interesting manuscript which dissects the release and impact of ACh in the retinal neuronal network. Previous work has highlighted the "local paracrine" actions of ACh in this highly specialized circuitry. The authors work significantly extends that of others and they should be congratulated. The use of a ACh 3 as an optical reporter of ACh release is particularly elegant. In general the experimental work seems to be well executed. The authors have taken pains to demonstrate some correlation between their functional work and high-resolution EM analysis of the structure. In the retinal field I think this paper is of high impact. My main concern is one of presentation, I in no way intend to prescribe to the authors. I however suggest that the particular circuitry studied is very specialized, ie dendro-dendritic contacts (close sites), action potential independent release etc, as the authors are keenly aware. Therefore, I think it is speculative to make general claims about a new mode of transmission based on this circuitry. I would prefer that the authors, clearly significant, findings are placed more into the context of the existing literature on the proposed actions of ACh in the retina. If the authors wish to make a more general argument they should also explore a tractable cholinergic system, such as that shown by Hestrin and Luthi in the neocortex, where fast "synaptic" ACh transmission occurs.

Reviewer #3 (Remarks to the Author):

The circuit that performs the computation of direction selectivity (DS) is one of the more understood circuits in the retina. DS computation begins in the dendrites of starburst amacrine cells (SACs), which preferentially respond to visual stimulation that progresses from the cell body to the periphery (centrifugal) as compared to visual motion of the opposite direction (centripetal). SACs make synaptic contacts with, and release GABA and ACh onto DS ganglion cells (DSGCs). Previous studies had shown that DSGCs are contacted by SAC dendrites whose preferred direction of motion (i.e., centrifugal motion) aligns with the DSGC's non-preferred (null) direction of motion. This asymmetrical connectivity results in a strong inhibition (release of GABA) for visual motion in the DSGC's null direction, but not for motion in the cell's preferred direction.

Interestingly, cholinergic neurotransmission between SACs and DSGCs does not appear to be selective for the direction of visual motion. Previous studies suggested that cholinergic input to DSGCs is either similar in magnitude for all directions or is greatest in the preferred direction – the opposite of GABAergic neurotransmission! The current manuscript focuses on ACh release from SACs and examines the synaptic properties of the cholinergic input to DSGCs.

The authors report several novel findings. First, imaging of cholinergic release with Ach3.0 revealed that the cholinergic input to DSGC dendrites is formed by multiple independent sites located ~5µm from each other. Each site is directionally tuned, but their tuning is random and unrelated to the preferred-null axis of the DSGC and the average selectivity across all sites is close to zero. This finding contradicts the idea that cholinergic neurotransmission between SACs and DSGCs operates via volume transmission as this form of neurotransmission is not expected to be DS at individual sites. Second, as would be expected of a chemical synapse, the authors demonstrate the presence of spontaneous miniature excitatory post-synaptic currents in DSGCs (minies). Interestingly, DSGCs with overlapping dendrites have correlated cholinergic minies. The authors interpret this finding as an indication of common cholinergic input. Third, dendrites of neighboring DSGCs co-fasciculate, and

in many cases, touch.

Based on these findings, the authors propose a new mechanism for synaptic release, which they call 'multi-directed' transmission. According to their model, cholinergic release sites are located on SAC varicosities that engulf null-DSGC dendrites signal directly to null dendrites but activate postsynaptic preferred dendrites that are up to 1um away.

While I find some of the aspects of the paper exciting and informative, I am not sure that the evidence that is provided support their conclusion. Below are my comments:

1. A. The authors mention that they use a cocktail of glutamate and GABA receptor blockers. What blockers have they used and at what concentration? Were these blockers used for the experiments shown in figure 2? Figure S2? The lack of clarity precludes an accurate interpretation of the experiments. If these blockers were used in all experiments, how could the authors produce GABAergic sIPSCs (figure S2)? If the blockers were not used, how can the authors distinguish between cholinergic and glutamatergic transmission? DSCGs are likely to be innervated by the same bipolar cells. If the glutamatergic transmission is not blocked, the finding that miniature EPSCs from distinct DSGCs are correlated could be attributed to shared bipolar cell inputs as opposed to shared cholinergic input.

B. What are the criteria for determining if two miniature EPSCs are synchronized? This detail is absent from the description of the methods.

C. Why was the data presented in Fig 2d normalized by the mean of all responses? Wouldn't it be better to show the correlation of the actual parameters?

2. It's likely that the percent of synchronized spontaneous EPSCs depends on the degree of dendritic overlap between DSGCs such that the more overlap between a pair DSGCs, the more likely they are to share the same cholinergic input. When using the percent of synchronized mEPSCs to estimate how far ACh spreads from its point of release, the authors report that 12% of the sEPSCs recorded from pairs of DSGCs occurred synchronously (pg. 7), but this measurement doesn't take in to account the degree of dendritic overlap between the pair of cells. For example, if the overlap in the dendritic fields is 12% and 12% of the sEPSCs recorded are coincident, then 100%, not 12% of cholinergic synapses that can be shared are in fact, synchronized! To more accurately describe the amount of shared cholinergic input between a pair of DSGCs, the amount of dendritic overlap should be calculated for each cell and compared to the rate of coincidental events. This normalization procedure has the potential to change the conclusions described in figure 3.

3. The description of the nearest neighbor analysis is significantly lacking. What software was used? What was the length of each dendritic segment that was used in the analysis? Based on their imaging results, individual cholinergic sites are spaced 5-10um apart. Were the dendrites divided into 5-10um long segments and nearest neighbor distance computed for each segment? Also, only a fraction of SAC varicosities seems to be contributing to functional release (Fig. 4b). How does this information affect the nearest neighbor analysis?

4. Given the points raised in (2,3), can the fraction of touching DSGCs dendrites explain the correlations observed in sEPSCs? I am surprised that the authors did not investigate the scenario in which a conventional cholinergic synapse co-activates two postsynaptic release sites, similar to the release from ribbon synapses. In this scenario, cholinergic synapses are distributed on SAC dendrites at sites of contacts between SACs and 2 DSGC dendrites (tripartite complexes, fig. 3). Ach release activates both dendrites, and because the postsynaptic dendrites seem to be more likely to be oriented in opposite directions (Fig. 3e), the total cholinergic release is non-DS. This scenario can explain the focal activation sites resolved in the imaging experiments without involving diffuse transmission.

5. I thought the framing of the paper was a little too general. The research is specific to the retina,

and people who study retina are the intended audience, but the authors frame the paper as being an investigation of cholinergic neurotransmission in general. I'm not sure that the results of these experiments are generalizable to cholinergic neurotransmission in other brain areas, and I don't think the manuscript should frame this research as such if there is no explicit comparison to cholinergic neurotransmission in other brain areas. The authors could frame the research as a more specific investigation of how cholinergic neurotransmission operates on the post-synaptic dendrites of DSGCs located on the preferred side of the cell and where it comes from.

6. The introduction and Figure 1 could be more streamlined. I think that early on in the discussion of the results (when they are referencing their models in figure 1), the authors should emphasize that chemical synapses are made between SACs and DSGCs only on the dendrites of the null side of the DSGC. Without this information, it's hard to follow their logic when considering each possible mode of ACh neurotransmission. I would also highlight the big question that the paper addresses – what is the origin of cholinergic input to preferred DSCG dendrites.

7. In figure 1 c-f, I would appreciate if the authors labeled the components of the synapse indicated graphically. It took me longer than it should have to understand each model because of the time spent trying to figure out what each of these components was meant to represent.

8. I think the result presented in figure 3f was interesting and worthy of discussion, but the authors spend very little time discussing it other than to say the absence of pairs of DSGCs participating in a tripartite synapse with the same directional preference is an exception to the finding that there is no relationship between the directional preference of the pairs of cells. Shouldn't more emphasis be placed on this result? It shows that there is some kind of relationship between the preferred direction of the DSGC contacted via a wrap-around synapse and the preferred direction of the DSGC with a dendrite proximal to the synapse.

9. The transition from results to the discussion is very harsh and unexpected, and the discussion itself is very short.

10. I couldn't see the purpose of the information presented in figure S5. There are a lot of colors. I assume that the light blue arrows (note the typo on figure legend) are important. But I couldn't see them.

11. What is the purpose of figure S8 a,b?

REVIEWER COMMENTS

Reviewer #1 (Remarks to the Author):

The authors set out to determine the mode of cholinergic synapses between starburst amacrine cells and direction selective retinal ganglion cells. They elegantly combine several methods, including electrophysiology, anatomical reconstruction, and 2-photon imaging of ACh in order to compare different models of direct and diffuse synaptic transmission. The mechanism of cholinergic transmission and its role in establishing direction selectivity in the retina are active questions in the field, and this study is timely and relevant. I support publication of this paper with some revisions to improve the clarity of the manuscript.

My primary concern was that I found the manuscript a bit difficult to read which I think this stems from the fact that there are many different techniques, and most readers will not have equal experience with, or understanding of, each of them. I think the manuscript would be greatly improved by putting just a bit more explanation and unpacking into the descriptions of the techniques and key analyses.

We're glad the Reviewer shares our enthusiasm for this study. To improve the clarity we have now distributed the results over 7 figures (rather than 4 figures), which we hope will clarify the logical flow and better describe the different techniques used.

1) Figure 1 c-f. These schematics are extremely important and valuable. However, a bit more explanation would help. What exactly is the key difference between “spillover” (d) and “multi-directed” (f)? I assume it's related to the position of the receptor in the schematic, but I don't quite understand why that yields the different traces shown.

This is an insightful question, which aims at the heart of this study. Thank you for bringing it up.

Spillover transmission is usually not associated with quantal responses. The levels of transmitter spilling over from neighbouring synapses are thought to generate weak slow responses that normally fall below the detection limit of whole-cell recordings. During multi-directed transmission, ACh release from individual vesicles generates synchronous activity at proximal and distal postsynaptic sites. Strikingly, the response amplitude and kinetics are comparable at both sites.

We now more carefully detail these differences between “spillover” and “multi-directed” transmission in a separate section. In addition, to formalize our intuition regarding the underlying mechanisms, we entered into a collaboration with Dr. Grosman, who is a world leader in nAChR biophysics. We present a new computational model that demonstrates how proximal and distal AChRs can respond similarly to a pulse of ACh release from a single vesicle, based on the

theoretical diffusion equations and detailed biophysical properties of nAChRs (Supplementary Fig. 5).

2) The frequency of correlated sEPSCs (p. 5) – the authors state this is more than expected given the spontaneous firing rates of the cells. This seems right to me, but I'd like this logic unpacked, and assumptions either demonstrated or at least stated.

In the revised manuscript, we state “*The low rate of sEPSCs (~3Hz) made the random occurrence of synchronized sEPSCs arising from independent sources unlikely (probability 0.000009)*”.

3) The fact that correlated decay kinetics are indicative of ACh concentration profiles rather than deactivation (on p. 6) – why is this? Please unpack the logic for this.

We have removed this statement from the revised manuscript.

4) P7 “the fraction of dendrites of a given DSGC, that were within a certain distance of the dendrites of a second DSGC, grew rapidly as this distance increased”. This sentence was very confusing to me and a more careful explanation would help a lot. I would also ask for an intuition for a null hypothesis. The assertion that it “grew rapidly” suggests that there’s an assumption that it should either not grow or grow slowly? (perhaps minor detail, but I also don’t think “rapidly” is the correct word here as it implies a temporal component rather than spatial)

We agree with the Reviewer, and have removed the word rapidly. To provide a better intuition for the analysis we now present two example pairs of DSGCs with different levels of dendritic overlap. We explicitly point out how the NNDs for these pairs change more or less sharply as a function of distance, depending on the degree of their dendritic overlap (Page 7).

5) Figure 3c: this appears to be a crucial analysis that ties together the different pieces. It is not clear to me what the p-values inset in the panels are nor how this supports the conclusion regarding the strength of correlation and distance. My understanding is that the correlation values (not shown) are too high at 0 μm , and too low at 2 and 3 μm , but close at 1 μm ? Again, I think this can be resolved through just a bit more unpacking in the text.

The Reviewers understand this correctly. The p-values are for paired t-tests evaluates whether the nearest neighbour distance distribution (over a given distance) is similar to the distribution of sEPSC correlation strengths. For NNDs at 0 μm , the NNDs were significantly lower than the sEPSC correlation strength ($p = 0.002$), while the NNDs <2 or 3 μm were significantly higher

than the sEPSC correlations ($p = 0.002, 0.0001$). Only for the NND distribution $<1 \mu\text{m}$ was the distribution not significantly different from the distribution of the sEPSC correlations ($p = 0.062$). This analysis was first performed by Trong and Rieke (2008) whom we cite.

6) A few of the supplementary figures seemed like they should be incorporated into the primary figures as they provided key evidence for the questions being addressed. Sup Figures 2 and 4 in particular.

We incorporated Supplementary Fig. 2 into Fig. 2 in the main text. The discussion on decay kinetics of cholinergic sEPSCs and associated Supplementary Fig. 4 have been removed from the current version of the manuscript.

7) Is there a citation for the fact that Oxt-T2A-Cre labels nasal on-off DS-RGCs? Or is this a new detail regarding this Cre line that has not been shown – in which case please add a Supplemental Figure that documents this more. Eg. how selective is its expression among RGC types?

As suggested, we have added a supplementary figure in which we better characterized the newly identified Oxt-T2A-Cre x Thy1-STOP-EYFP line (new **Supplementary Fig. 7**). Spike recordings from labeled cells with large somas revealed that they all belonged to the class of nasally tuned ON-OFF DSGCs ($n=13/13$). We also show that this line also labels a few starbursts.

As noted in the text (pg. 11), ACh signals were measured in DSGC dendrites that were filled with a red dye through the patch electrode to ensure signals were measured from DSGCs, and not starbursts.

Reviewer #2 (Remarks to the Author):

This very interesting manuscripts which dissects the release and impact of ACh in the retinal neuronal network. Previous work has highlighted the “local paracrine” actions of ACh in this highly specialized circuitry. The authors work significantly extends that of others and they should be congratulated. The use of a ACh 3 as an optical reporter of ACh release is particularly elegant. In general the experimental work seems to be well executed. The authors have taken pains to demonstrate some correlation between their functional work and high-resolution EM analysis of the structure. In the retinal field I think this paper is of high impact. My main concern is one of presentation, I in no way intend to prescribe to the authors. I however suggest that the particular circuitry studied is very specialized, ie dendro-dendritic contacts (close sites), action potential independent release etc, as the authors are keenly aware. Therefore, I think it is speculative to make general claims about a new mode of transmission based on this circuitry. I

would prefer that the authors, clearly significant, findings are placed more into the context of the existing literature on the proposed actions of ACh in the retina. If the authors wish to make a more general argument they should also explore a tractable cholinergic system, such as that shown by Hestrin and Luthi in the neocortex, where fast “synaptic” ACh transmission occurs.

We thank the reviewer for their encouraging words. We now re-written the manuscript framing the results in the context of the DS circuit.

Reviewer #3 (Remarks to the Author):

While I find some of the aspects of the paper exciting and informative, I am not sure that the evidence that is provided support their conclusion. Below are my comments:

1. A. The authors mention that they use a cocktail of glutamate and GABA receptor blockers. What blockers have they used and at what concentration? Were these blockers used for the experiments shown in figure 2? Figure S2? The lack of clarity precludes an accurate interpretation of the experiments. If these blockers were used in all experiments, how could the authors produce GABAergic sIPSCs (figure S2)? If the blockers were not used, how can the authors distinguish between cholinergic and glutamatergic transmission? DSCGs are likely to be innervated by the same bipolar cells. If the glutamatergic transmission is not blocked, the finding that miniature EPSCs from distinct DSGCs are correlated could be attributed to shared bipolar cell inputs as opposed to shared cholinergic input.

In all the electrophysiological experiments shown in the study, glutamatergic transmission was blocked with the following pharmacological agents: 50 μ M DL-AP4 (NMDA receptor antagonist and mGluR6 agonist), 20 μ M CNQX (AMPA/Kainate receptor antagonist) and 100 μ M UBP-310 (Kainate receptor antagonist). Hence, it is unlikely that the shared input is mediated by glutamate from bipolar cells. These details have been added to the text (pg. 5).

To further strengthen this conclusion, we have included new data showing that these spontaneous events are blocked by the non- α 7 nAChR antagonist (DHBE) (**Supplementary Fig. 2**). In this dataset, in addition to blocking glutamatergic transmission, the activity of GABA_A receptors was also blocked with 10 μ M SR-95531.

B. What are the criteria for determining if two miniature EPSCs are synchronized? This detail is absent from the description of the methods.

We measured the sEPSC triggered average (STA) currents, where currents were averaged in a given DSGC over the same time period in which sEPSCs occurred in its neighbor.

C. Why was the data presented in Fig 2d normalized by the mean of all responses? Wouldn't it be better to show the correlation of the actual parameters?

Standardization was necessary to avoid correlation artefacts, which could be induced by the cell-to-cell variability in mean and standard deviation (especially for peak amplitude). This point is made in the legend of figure 2.

2. It's likely that the percent of synchronized spontaneous EPSCs depends on the degree of dendritic overlap between DSGCs such that the more overlap between a pair DSGCs, the more likely they are to share the same cholinergic input.

The Reviewer's intuition is correct. However, if the dendritic overlap is estimated coarsely (for example by measuring the fractional area of dendritic overlap; figure on right) then the degree of overlap does not relate to the correlation strength). It is only when the dendritic overlap is considered at a finer spatial scale are they found to be predictive of the correlation strength (as detailed in response to Reviewer 1: point 5)

When using the percent of synchronized mEPSCs to estimate how far ACh spreads from its point of release, the authors report that 12% of the sEPSCs recorded from pairs of DSGCs occurred synchronously (pg. 7), but this measurement doesn't take in to account the degree of dendritic overlap between the pair of cells. For example, if the overlap in the dendritic fields is 12% and 12% of the sEPSCs recorded are coincident, then 100%, not 12% of cholinergic synapses that can be shared are in fact, synchronized! To more accurately describe the amount of shared cholinergic input between a pair of DSGCs, the amount of dendritic overlap should be calculated for each cell and compared to the rate of coincidental events. This normalization procedure has the potential to change the conclusions described in figure 3.

We agree with the Reviewer's notion that most of the cholinergic transmission sites that can be shared are in fact synchronized. We now explicitly state this in the revised manuscript in pg. 8.

“The close correspondence between the dendritic overlap over the $<1 \mu\text{m}$ scale and coincident sEPSCs indicates that most starburst inputs that can be shared, in fact produce synchronized postsynaptic responses, supporting our anatomical findings (Fig. 1).”

3. The description of the nearest neighbor analysis is significantly lacking. What software was

used? What was the length of each dendritic segment that was used in the analysis? Based on their imaging results, individual cholinergic sites are spaced 5-10um apart. Were the dendrites divided into 5-10um long segments and nearest neighbor distance computed for each segment?

This point has been previously addressed (See Reviewer 1:point 5).

Also, only a fraction of SAC varicosities seems to be contributing to functional release (Fig. 4b). How does this information affect the nearest neighbor analysis?

The fraction of starburst varicosities releasing ACh will determine the total cholinergic inputs to DSGCs. However, the nearest neighbor analysis was performed using sEPSC correlation strength, which is a measure that is normalized to the number of cholinergic inputs. Hence, the nearest neighbour analysis is not be affected by the fraction of varicosities releasing ACh.

4. Given the points raised in (2,3), can the fraction of touching DSGCs dendrites explain the correlations observed in sEPSCs? I am surprised that the authors did not investigate the scenario in which a conventional cholinergic synapse co-activates two postsynaptic release sites, similar to the release from ribbon synapses. In this scenario, cholinergic synapses are distributed on SAC dendrites at sites of contacts between SACs and 2 DSGC dendrites (tripartite complexes, fig. 3). Ach release activates both dendrites, and because the postsynaptic dendrites seem to be more likely to be oriented in opposite directions (Fig. 3e), the total cholinergic release is non-DS. This scenario can explain the focal activation sites resolved in the imaging experiments without involving diffuse transmission.

We are in agreement with the Reviewer on this issue, and in fact this was our main hypothesis in the previous version that appears to have gone amiss. In attempt to avoid future misunderstandings, we now first present SBEM evidence for a tripartite synapse (presynaptic terminal juxtaposing two postsynaptic elements) and more clearly state throughout the text that the results support a model in which ACh release from a single terminal spreads to activate receptors on both postsynaptic elements (similar to the photoreceptor synapse).

We also include an in-depth discussion outlining the similarities and differences between multi-directed transmission and “engineered spillover” at the photoreceptor synapse (Sterling and Matthews, 2005).

5. I thought the framing of the paper was a little too general. The research is specific to the retina, and people who study retina are the intended audience, but the authors frame the paper as being an investigation of cholinergic neurotransmission in general. I'm not sure that the results of these experiments are generalizable to cholinergic neurotransmission in other brain areas, and I don't think the manuscript should frame this research as such if there is no explicit

comparison to cholinergic neurotransmission in other brain areas. The authors could frame the research as a more specific investigation of how cholinergic neurotransmission operates on the post-synaptic dendrites of DSGCs located on the preferred side of the cell and where it comes from.

We agree. We now frame the results in the context of the DS circuitry.

6. The introduction and Figure 1 could be more streamlined. I think that early on in the discussion of the results (when they are referencing their models in figure 1), the authors should emphasize that chemical synapses are made between SACs and DSGCs only on the dendrites of the null side of the DSGC. Without this information, it's hard to follow their logic when considering each possible mode of ACh neurotransmission. I would also highlight the big question that the paper addresses – what is the origin of cholinergic input to preferred DSCG dendrites.

Agreed. We now start by describing the circuitry in the introduction (pg. 2-3), and emphasize the mismatch between the functional patterns of cholinergic connectivity and the anatomical wiring.

7. In figure 1 c-f, I would appreciate if the authors labeled the components of the synapse indicated graphically. It took me longer than it should have to understand each model because of the time spent trying to figure out what each of these components was meant to represent.

We apologize for this. The figure has been now been updated with the suggested changes (Fig. 4) and associated with Supplementary Fig. 5, which uses a model to supports the ideas in the schematic.

8. I think the result presented in figure 3f was interesting and worthy of discussion, but the authors spend very little time discussing it other than to say the absence of pairs of DSGCs participating in a tripartite synapse with the same directional preference is an exception to the finding that there is no relationship between the directional preference of the pairs of cells. Shouldn't more emphasis be placed on this result? It shows that there is some kind of relationship between the preferred direction of the DSGC contacted via a wrap-around synapse and the preferred direction of the DSGC with a dendrite proximal to the synapse.

The lack of secondary contacts from DSGCs coding the same direction is not surprising as DSGCs encoding the same direction tend to have less dendritic overlap. We have noted this in the results section (pg. 5)

9. The transition from results to the discussion is very harsh and unexpected, and the discussion itself is very short.

We have reworked this section and now discuss in detail how:

- discrepancies between functional and anatomical connectivity in the starburst-DSGC network (pg. 12-14) can be reconciled by our new results.
- ‘multi-directed’ transmission is different from previously described forms of chemical transmission (pg. 12-14).
- the [ACh] cleft and biophysical properties of the nAChRs help shape multi-directed transmission (pg. 15).
- our results impact the DSGC’s direction selectivity (pg. 16)
- our results shape how we think about cholinergic transmission in other parts of the central nervous system (pg. 17-18).

10. I couldn't see the purpose of the information presented in figure S5. There are a lot of colors. I assume that the light blue arrows (note the typo on figure legend) are important. But I couldn't see them.

The aim of this figure was to illustrate how the directional preference of the DSGCs shown in Fig. 1 was determined from the anatomy alone. We have modified both the main figure and the supplement to make this more clear (See Fig. 1c and Supplementary Fig. 1b).

11. What is the purpose of figure S8 a,b?

See response to Reviewer 1: point 7

Reviewer #1 (Remarks to the Author):

The authors have improved the clarity of the manuscript considerably, and have addressed my primary concerns. I note a few locations where I think small additions to figure legends or text can improve clarity just a bit more.

The Results section describing Figure 2, under heading “Rapid, multi-directed cholinergic transmission,” will confuse readers. The authors first describe the miniature events, for which the fact that these are paired recordings isn’t important, before describing the synchronicity of the events. However, we readers look at the figure and read the figure legends, the first thing that they take in is that these are paired recordings. I don’t know if it makes sense to add a panel to the figure about the events on their own, or to rework the text to make this section flow more logically, but I did find this problematic.

Fig 2d, what is the distinction between the two blue traces? My understanding from the legend is that these are the averages that result from considering each of the paired DSGCs as the reference cell, but it is unclear to me why that should yield such different traces.

Fig 4d, what is Pr?

Fig 6c the dotted line is too faint. Please make this more visible

In the discussion, the authors make the point that ACh is insensitive to the asymmetric distribution of wraparound synapses. This makes me wonder why there is an asymmetric distribution wraparound synapses. I don’t imagine there is an easy answer to this, but I think it might be worth speculating or even just raising the question.

Reviewer #2 (Remarks to the Author):

The revised manuscript is well crafted and addresses the issue of focus that I previously raised. In general I find that the data is well illustrated and described. I am slightly concerned by the implication in the abstract that the authors used serial block em techniques, when in fact they analysed a previously obtained data set (ref 32). I think this should be corrected. I very much support publication.

Reviewer #3 (Remarks to the Author):

I enjoyed reading the revised version of the manuscript. I think that the authors have sufficiently addressed my comments (and the comments of the other reviewers).

Reviewer #1 (Remarks to the Author):

The authors have improved the clarity of the manuscript considerably, and have addressed my primary concerns. I note a few locations where I think small additions to figure legends or text can improve clarity just a bit more.

The Results section describing Figure 2, under heading “Rapid, multi-directed cholinergic transmission,” will confuse readers. The authors first describe the miniature events, for which the fact that these are paired recordings isn’t important, before describing the synchronicity of the events. However, we readers look at the figure and read the figure legends, the first thing that they take in is that these are paired recordings. I don’t know if it makes sense to add a panel to the figure about the events on their own, or to rework the text to make this section flow more logically, but I did find this problematic.

We agree with the Reviewer and have modified this section as follows:

*“Dual voltage-clamp recordings from nearby DSGCs (inter-somatic distance < 50µm; **Fig. 2a**) revealed that a significant fraction of spontaneous cholinergic miniature excitatory postsynaptic currents (sEPSCs) occurred synchronously (**Fig. 2b-d**). Spontaneous inputs were measured at a holding potential of -60 mV (~ E_{Cl}) in the presence of a drug cocktail containing 50µM DL-AP4, 20µM CNQX and 100µM UBP-310, which block AMPA, NMDA and kainate-type glutamate receptors, as well as ON bipolar cell activity mediated by type 6 metabotropic glutamate receptors. Further pharmacological analysis revealed that sEPSCs are largely mediated by $\alpha 6$ subunit-containing receptors ($n=5$; **Supplementary Fig. 2**), consistent with single cell transcriptomic profiling results that suggest a predominant expression of $\alpha 6$ -nAChRs in DSGCs³⁶.”*

Fig 2d, what is the distinction between the two blue traces? My understanding from the legend is that these are the averages that result from considering each of the paired DSGCs as the reference cell, but it is unclear to me why that should yield such different traces.

-Thank you for pointing this out. The light blue line shows the average STA normalized to the sEPSC. We have added this statement to the figure legend.

Fig 4d, what is Pr?

Pearson’s correlation coefficient. We have changed it to the standard symbol (r).

Fig 6c the dotted line is too faint. Please make this more visible

Changed to a solid black contour.

In the discussion, the authors make the point that ACh is insensitive to the asymmetric

distribution of wraparound synapses. This makes me wonder why there is an asymmetric distribution wraparound synapses. I don't imagine there is an easy answer to this, but I think it might be worth speculating or even just raising the question.

The asymmetry in the distribution of wraparound synapses is the basis for asymmetric GABAergic inhibition, which is critical for shaping the directional responses of DSGCs. This is stated in the introduction (pg 2-3).

*“In a tour de force, Briggman et al. (2011) reconstructed the starburst-DSGC circuitry using serial block-face electron microscopy (SBEM), and observed a prominent asymmetry in their synaptic connectivity²⁵ (**Fig. 1b**). They noted that the majority (~90%) of the large ‘wraparound’ synaptic contacts made by radiating starburst dendrites arise from cells that have their somas displaced towards the ‘null-side’ of the DSGC’s receptive field (i.e., the side from which null stimuli first enter the receptive field; **Fig. 1b**). These ‘null-connections’ provide the structural basis for the asymmetric GABAergic inhibition that confers direction selectivity to DSGCs²⁷⁻³² (**Fig. 1b**).”*

Reviewer #2 (Remarks to the Author):

The revised manuscript is well crafted and addresses the issue of focus that I previously raised. In general I find that the data is well illustrated and described. I am slightly concerned by the implication in the abstract that the authors used serial block em techniques, when in fact they analysed a previously obtained data set (ref 32). I think this should be corrected. I very much support publication.

In response to the Reviewer’s concerns we now state that we analysed a SBEM dataset and reference the paper by Ding et al., “Analysis of an SBEM data set²⁶ revealed....”

Reviewer #3 (Remarks to the Author):

I enjoyed reading the revised version of the manuscript. I think that the authors have sufficiently addressed my comments (and the comments of the other reviewers).

Once again, we thank the Reviewer for the critical comments.